# An intrinsic mechanism of metabolic tuning promotes cardiac resilience to stress

Matteo Sorge [1✉], Giulia Savoré [1], Andrea Gallo [1], Davide Acquarone [1], Mauro Sbroggiò[1], Silvia Velasco [1], Federica Zamporlini[2], Saveria Femminò[3], Enrico Moiso[1], Giampaolo Morciano[4,5], Elisa Balmas[1], Andrea Raimondi[6], Gabrielle Nattenberg [7], Rachele Stefania[1], Carlo Tacchetti [6], Angela Maria Rizzo [8], Paola Corsetto[8], Alessandra Ghigo [1], Emilia Turco[1], Fiorella Altruda[1], Lorenzo Silengo[1], Paolo Pinton [4,5], Nadia Raffaelli[2], Nathan J Sniadecki[7], Claudia Penna[3], Pasquale Pagliaro[3], Emilio Hirsch[1], Chiara Riganti[9], Guido Tarone[1,10], Alessandro Bertero [1,11] & Mara Brancaccio [1,11✉]

## Abstract

**Defining the molecular mechanisms underlying cardiac resilience is crucial to find effective approaches to protect the heart. A physiologic level of ROS is produced in the heart by fatty acid oxidation, but stressful events can boost ROS and cause mitochondrial dysfunction and cardiac functional impairment. Melusin is a muscle specific chaperone required for myocardial compensatory remodeling during stress. Here we report that Melusin localizes in mitochondria where it binds the mitochondrial trifunctional protein, a key enzyme in fatty acid oxidation, and decreases it activity. Studying both mice and human induced pluripotent stem cell-derived cardiomyocytes, we found that Melusin reduces lipid oxidation in the myocardium and limits ROS generation in steady state and during pressure overload and doxorubicin treatment, preventing mitochondrial dysfunction. Accordingly, the treatment with the lipid oxidation inhibitor Trimetazidine concomitantly with stressful stimuli limits ROS accumulation and prevents long-term heart dysfunction. These findings disclose a physiologic mechanism of metabolic regulation in the heart and demonstrate that a timely restriction of lipid metabolism represents a potential therapeutic strategy to improve cardiac resilience to stress.**

**Keywords** Cardiac Metabolism; ROS; Doxorubicin; Pressure Overload; Chaperone Proteins
**Subject Categories** Cardiovascular System; Metabolism

## Introduction

Despite possessing extensive metabolic flexibility, the human heart normally relies on long-chain fatty acid (LCFA) catabolism through mitochondrial β-oxidation (FAO) to produce ~85% of the ATP required for its effective and lifelong rhythmic contractile activity (Robson, 2021). A central hub in FAO is the mitochondrial trifunctional protein (MTP), a holoenzyme consisting of four α and four β subunits, catalyzing the last three reactions of LCFA β-oxidation. MTP is critically required for heart function, so that its deficiency leads to severe autosomal recessive cardiomyopathy (Dagher et al, 2021).

The cardiac metabolic preference for FAO, as defined by the Randle cycle (Hue and Taegtmeyer, 2009), assures that, when available, the myocardium preferentially oxidizes lipids instead of glucose to produce more ATP. However, this comes at the cost of increased ROS production. FAO generates ROS (Isei et al, 2022; Speijer et al, 2014; St-Pierre et al, 2002) and additional oxidative stress arises from glucose that, not being oxidized in the mitochondria, is directed toward other catabolic pathways, such as the polyol pathway and the hexosamine biosynthesis pathway (Battault et al, 2020; Mylonas et al, 2023). In addition, stressful situations, such as pressure overload (Nickel et al, 2015; Suetomi et al, 2018; Wang et al, 2018), ischemia/reperfusion (Bugger and Pfeil, 2020), or doxorubicin-induced cardiotoxicity (Berthiaume and Wallace, 2007), generate oxidative stress that further raises myocardial ROS levels above a certain threshold, triggering a self-amplification mechanism known as ROS-induced ROS release (RIRR) (Dey et al, 2018; Leach et al, 2001; Tse et al, 2016; Tullio et al, 2013; Zorov et al, 2000). RIRR further promotes ROS production from cytoplasmic sources, such as NADPH oxidases,

[1]Department of Molecular Biotechnologies and Health Sciences, Molecular Biotechnology Center "Guido Tarone", University of Turin, Turin 10126, Italy. [2]Department of Agricultural, Food and Environmental Sciences, Polytechnic University of Marche, Ancona 60121, Italy. [3]Department of Clinical and Biological Sciences, University of Turin, Orbassano 10043, Italy. [4]Department of Medical Sciences, University of Ferrara, Ferrara 44121, Italy. [5]Maria Cecilia Hospital, GVM Care and Research, Cotignola 48033, Italy. [6]Experimental Imaging Centre, Istituto di Ricovero e Cura a Carattere Scientifico (IRCCS) San Raffaele Scientific Institute, Milan 20132, Italy. [7]Departments of Mechanical Engineering, Bioengineering, and Laboratory Medicine and Pathology, Institute for Stem Cell and Regenerative Medicine, and Center for Cardiovascular Biology, University of Washington, Seattle, WA 98109, USA. [8]Department of Pharmacological and Biomolecular Sciences, University of Milan, Milano 20133, Italy. [9]Department of Oncology, University of Turin, Torino 10126, Italy. [10]Deceased: Guido Tarone. [11]These authors contributed equally: Alessandro Bertero, Mara Brancaccio. ✉E-mail: matteo.sorge@unito.it; mara.brancaccio@unito.it

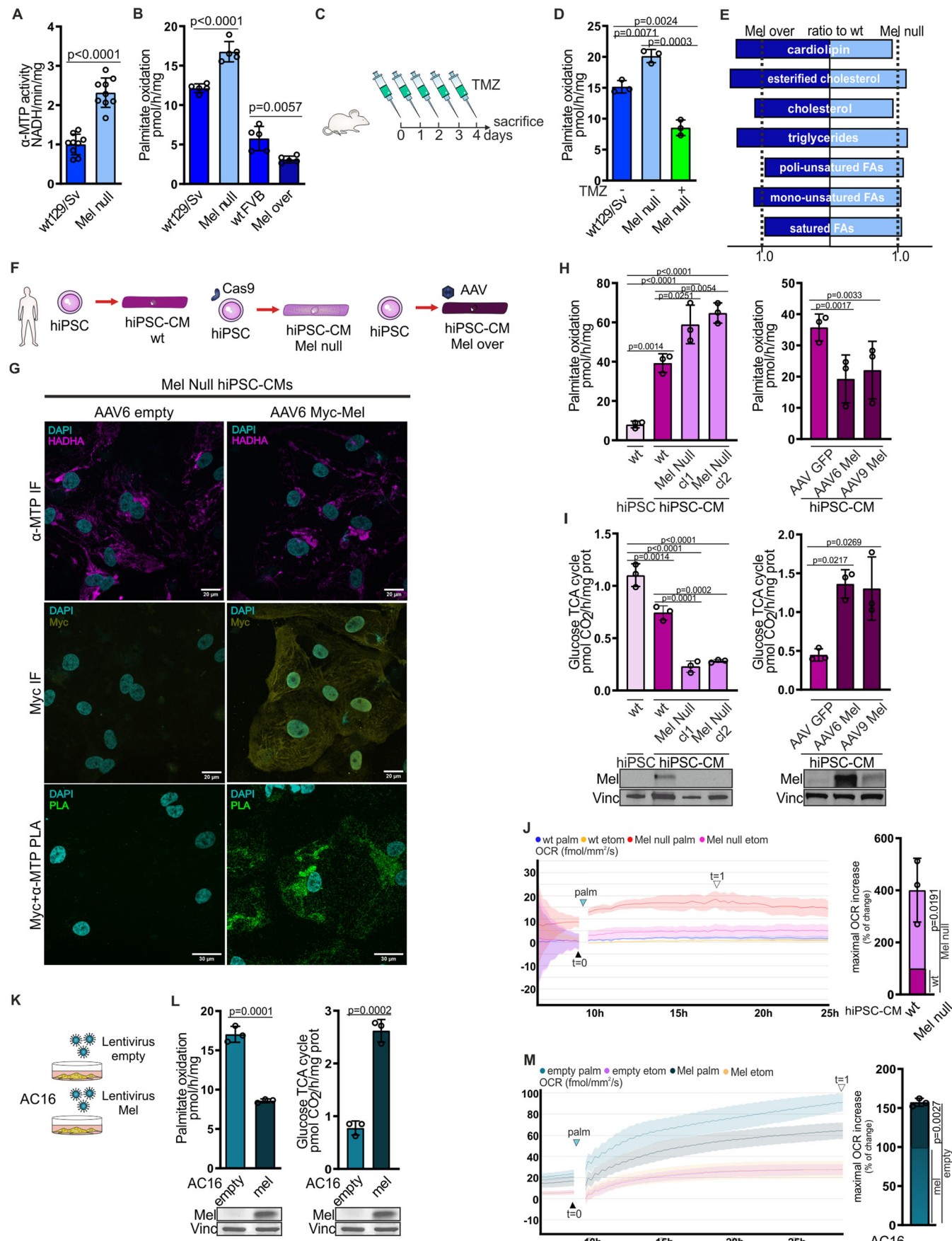

◄ **Figure 1. Melusin regulates FAO by inhibiting the mitochondrial trifunctional protein.**

(A) Modulation of α-MTP activity in wild-type and Mel null mice detected as NADH reduction in time in cardiac homogenates supplemented with acetoacetyl-CoA. ($n = 9$ per group; mean ± SD; $p$ value by unpaired $t$-test; Data normalized on wt129/Sv media set to 1). (B) Palmitate oxidation capacity of cardiac isolated mitochondria from Mel null, Mel over and wild-type mice evaluated as generation of radioactive metabolites of [1–14C]-palmitate ($n = 5$ per group; mean ± SD; $p$ value by unpaired $t$-test); (C) Trimetazidine treatment protocol: Mel null mice were treated orally from day 0 to day 4 with 30 mg/kg TMZ (TMZ+) or saline (TMZ−) as control. Wild-type mice were treated similarly with saline only. Mice were sacrificed at the end of the protocol to collect the hearts. (D) Palmitate oxidation capacity of cardiac isolated mitochondria from Mel null and wild-type mice treated as described in (C). ($n = 3$ per group; mean ± SD; $p$ value by two-way ANOVA with Bonferroni correction). (E) The ratio of the quantity (μg/mg of tissue) of lipids in Mel null and Mel over mice compared to wild-type counterparts. Lipids have been quantified by lipidomic analysis from cardiac tissues. ($n = 3$ per group; mean ± SD; $p$ value by unpaired $t$-test). (F) hiPSC cell models: wild-type hiPSCs were differentiated in wild-type hiPSC-CMs. hiPSCs knockout for Melusin were generated through CRISPR Cas9 technology and differentiated in Mel null hiPSC-CMs. hiPSCs were differentiated in hiPSC-CMs and infected with AAV6 and AAV9 carrying myc-Melusin, to over-express Melusin, or control AAV6 and AAV9. (G) Representative Immunofluorescence (IF) and PLA signal of Myc-Melusin (Myc-tag antibody) and α-MTP (HADHA antibody) in Mel null hiPSC-CMs infected with an AAV6-empty or carrying the construct for Myc-Melusin. DAPI was used to stain nuclei. Images were captured by confocal microscopy. Scale bare = 20/30 μm. Representative of $n = 3$ independent experiments (H) Palmitate oxidation capacity of isolated mitochondria from wild-type hiPSCs, wild-type hiPSC-CMs, and clone 1 and clone 2 Mel null hiPSC-CMs (left) and from AAV GFP hiPSC-CMs, AAV6 Mel hiPSC-CMs and AAV9 Mel hiPSC-CMs (right). ($n = 3$ per group; mean ± SD; $p$ value by one-way ANOVA with Bonferroni correction). (I) Glucose flux through TCA cycle detected as marked $CO_2$ produced from 6[$^{14}$C]-glucose via the tricarboxylic acid cycle in wild-type hiPSCs, wild-type hiPSC-CMs and clone 1 and clone 2 Mel null hiPSC-CMs (left) and in AAV GFP hiPSC-CMs, AAV6 Mel hiPSC-CMs and AAV9 Mel hiPSC-CMs (right). ($n = 3$ per group; mean ± SD; $p$ value by one-way ANOVA with Bonferroni correction). A representative immunostaining for Melusin and vinculin levels in the cells used in (H) and (I) is reported below. (J) Representative curves (left) of the OCR modulation in wild-type (blue) and Mel null (red) hiPSC-CMs cultured in a media added with palmitate-BSA (azure arrow). Wild-type (yellow) and Mel null (purple) hiPSC-CMs cultured in the presence of the carnitine palmitoyltransferase-1 inhibitor etomoxir were used as negative controls. Maximal OCR increase (right) expressed as a percentage of Mel null hiPSC-CMs compared to the 100% of wt hiPSC-CMs. The maximal OCR increase is calculated as the OCR difference between $t = 1$ (empty arrow) and $t = 0$ (full arrow). (curves represent the media of 4–6 technical triplicates for each condition, biological $n = 3$ per group; mean ± SD; $p$ value by one sample $t$-test). (K) AC16 cell model: AC16 cells were infected with a lentivirus carrying Myc-Melusin construct to over-express Melusin or an empty lentivirus as control. (L) Palmitate oxidation capacity of isolated mitochondria (left) and glucose flux through TCA cycle (right) in AC16 cells empty or expressing Melusin. A representative immunostaining for Melusin and vinculin level is reported below. (left: $n = 3$ mitochondrial preparations from independent experiments per group; right: $n = 3$ cell wells per group in independent experiments; mean ± SD; $p$ value by unpaired $t$-test; the representative blot is common for left and right graph). (M) Representative curves (left) of the OCR modulation in empty (light green) and Mel (dark green) AC16 cells cultured in a media added with palmitate-BSA (azure arrow). Empty (violet) and Mel (yellow) AC16 cells cultured in the presence of etomoxir were used as negative controls. Maximal OCR increase (right) expressed as a percentage of empty AC16 cells compared to 100% of Mel AC16 cells. The maximal OCR increase is calculated as the OCR difference between $t = 1$ (empty arrow) and $t = 0$ (full arrow). (curves represent the media of 4–6 technical triplicates for each condition, biological $n = 3$ per group; mean ± SD; $p$ value by one sample $t$-test). TMZ trimetazidine, FAs fatty acids, Cas9 CRISPR Cas9 technology, PLA proximity ligation assay, TCA tricarboxylic acid; OCR oxygen consumption rate, palm palmitate-BSA, etom etomoxir. Source data are available online for this figure.

uncoupled nitric oxide synthase, and xanthine oxidase (Bertero and Maack, 2018; Daiber et al, 2017; Dey et al, 2018; Koju et al, 2019; Sun et al, 2020). When their production overwhelms the cardiac antioxidant defenses, ROS begin to damage macromolecules, membranes, and organelles. Indeed, excessive ROS may lead to mitochondrial dysfunction, promoting an energetic crisis that pushes the heart into failure (Aon et al, 2003; Dai et al, 2011; Dey et al, 2018).

It is also known that stressful events rapidly modify the cardiac metabolism, inducing a shift from fatty acids towards glucose utilization (Akki et al, 2008; Kolwicz et al, 2013; Stanley et al, 2005; Zhong et al, 2013). Whether this metabolic rewiring is beneficial, given that glucose is oxidized at a lower oxygen cost compared to fatty acids, or detrimental, due to the lower ATP compared to fatty acids, is still a matter of debate (van Bilsen et al, 2009). Over time, progressive mitochondrial dysfunction impairs oxidative metabolism and, thus, the heart becomes mainly dependent on glycolysis, generating a chronic energetic crisis that contributes to heart failure (Lopaschuk et al, 2021). Whether specific factors may tune cardiac metabolism to mitigate ROS accumulation and promote heart resilience is unclear.

Melusin has been characterized as a cytoplasmic muscle-specific chaperone protein with a cardioprotective role in a wide variety of heart insults, such as pressure overload, myocardial infarction, ischemia-reperfusion injury, and sepsis-induced cardiomyopathy (Arina et al, 2022; Brancaccio et al, 2003; De Acetis et al, 2005; Penna et al, 2014; Unsold et al, 2014). In particular, in a model of pressure overload involving mice subjected to transverse aortic constriction (TAC), the absence of Melusin impairs compensatory

remodeling of the left ventricle, leading to a deterioration of contractile function and a rapid transition toward heart dilation (Brancaccio et al, 2003). Conversely, the overexpression of Melusin in the heart of mice undergoing prolonged pressure overload induces the establishment of persistent compensatory remodeling and the preservation of the contractile function, protecting the myocardium against dilation and failure (De Acetis et al, 2005).

Melusin is already known to bind to the cytoplasmic region of β1 integrin (Brancaccio et al, 1999) and transduce mechanical signals through the activation of the AKT and ERK pathways, mediating cardiomyocytes survival and compensatory hypertrophy (Brancaccio et al, 2003; De Acetis et al, 2005). Here, we report that Melusin can also localize to the mitochondria, where it binds to MTP, modulating FAO and ROS generation in cardiomyocytes. We found that this role is essential when stressful stimuli push further the ROS level and protect the heart from mitochondrial dysfunction and oxidative stress. Indeed, our findings reveal that tuning cardiac metabolism in concomitance to stressful events promotes cardiac resilience to stress.

## Results

### Melusin regulates fatty acid metabolism by binding and inhibiting the mitochondrial trifunctional protein

Co-immunoprecipitation experiments coupled with mass spectrometry led us to identify the interaction of the muscle-specific chaperone Melusin with the α and β subunits of the mitochondrial

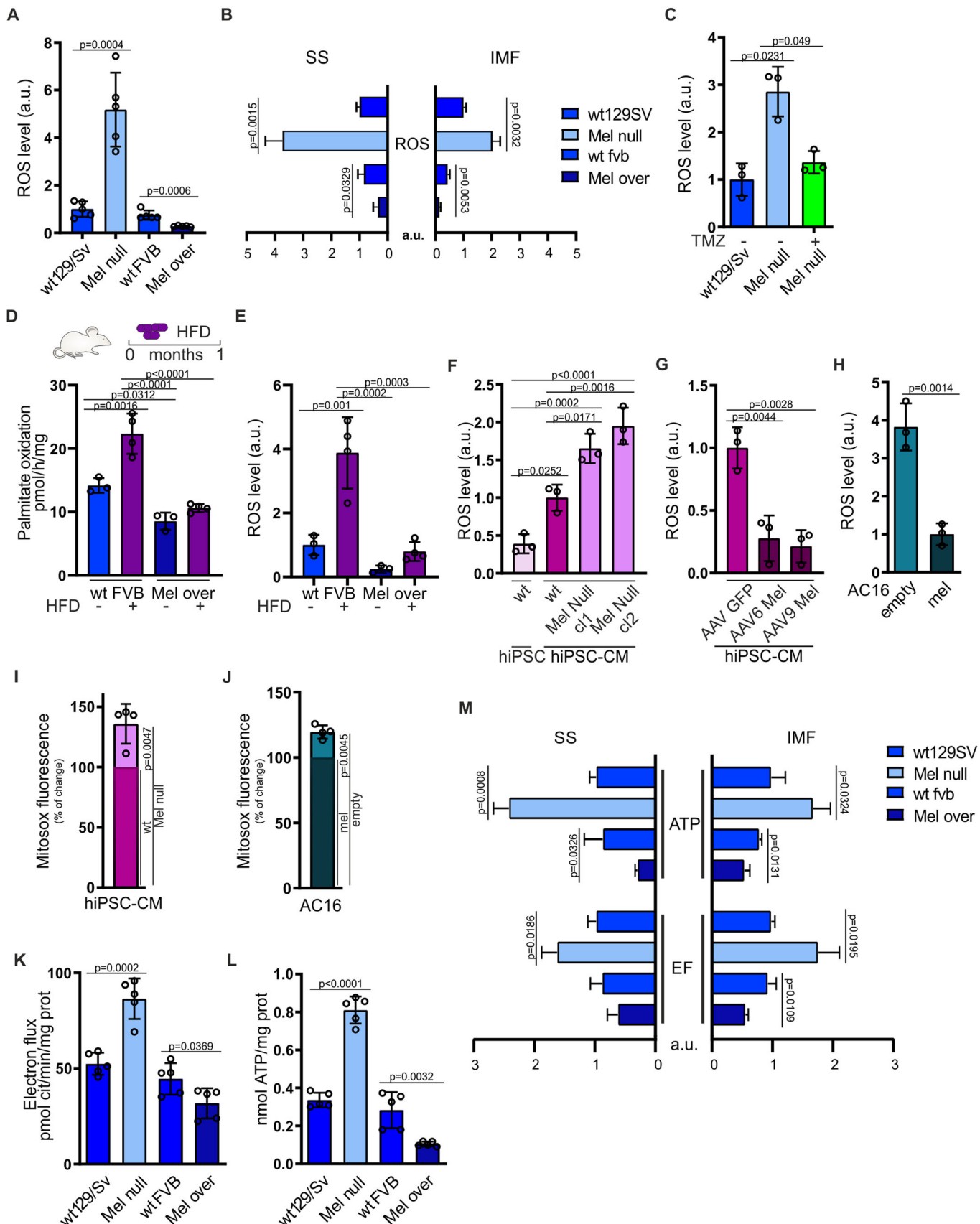

**Figure 2.  Melusin regulates ROS generation and mitochondrial function in the myocardium.**

(A) ROS content in Mel null, Mel over, and wild-type mice evaluated as DCFDA-AM fluorescence in isolated mitochondria from cardiac tissue. ($n = 5$ per group; mean ± SD; $p$ value by unpaired $t$-test; Data normalized on wt129/Sv media set to 1). (B) ROS content in subsarcolemmal (SS) and intermyofibrillar (IMF) mitochondria isolated from Mel null, Mel over, and wild-type hearts. ($n = 3$ per group; mean ± SD; $p$ value by unpaired $t$-test; Data normalized on wt129/Sv media set to 1). (C) ROS content in isolated mitochondria from cardiac tissue of wild-type and Mel null mice treated as described in 1C. ($n = 3$ per group; mean ± SD; $p$ value by two-way ANOVA with Bonferroni correction; Data normalized on wt129/Sv TMZ - media set to 1). (D) Modulation of palmitate oxidation estimated as radioactive metabolites of [1-$^{14}$C]-palmitate generated in cardiac isolated mitochondria from wild-type and Mel over mice fed with high-fat diet (HFD +) or normal diet (HFD−) for 5 weeks. ($n = 3$ per HFD −; $n = 4$ per HFD +; mean ± SD; $p$ value by two-way ANOVA with Bonferroni correction). (E) ROS content evaluated as DCFDA-AM fluorescence in cardiac isolated mitochondria from wild-type and Mel over mice feeded as in (D). ($n = 3$ per HFD −; $n = 4$ per HFD +; mean ± SD; $p$ value by two-way ANOVA with Bonferroni correction; Data normalized on wt FVB HFD - media set to 1). (F) ROS content in mitochondria from wild-type hiPSCs, wild-type hiPSC-CMs, and clone 1 and clone 2 Mel null hiPSC-CMs. ($n = 3$ per group; mean ± SD; $p$ value by one-way ANOVA with Bonferroni correction; Data normalized on wt hiPSC-CMs media set to 1). (G) ROS content in mitochondria from AAV GFP hiPSC-CMs, AAV6 Mel hiPSC-CMs, and AAV9 Mel hiPSC-CMs. ($n = 3$ mitochondrial preparations per group from hiPSC-CMs independently differentiated and infected; mean ± SD; $p$ value by one-way ANOVA with Bonferroni correction; Data normalized on AAV GFP media set to 1). (H) ROS content in mitochondria from AC16 empty and mel. ($n = 3$ mitochondrial preparations from independent experiments; mean ± SD; $p$ value by paired $t$-test; Data normalized on mel cells media set to 1). (I) Superoxide anion content was evaluated as MitoSOX fluorescence intensity per cell by confocal live imaging. Data expressed as a percentage of Mel null hiPSC-CMs compared to the 100% wt hiPSC-CMs. (technical $n = 3$ cell wells per condition, biological $n = 4$ independent experiments per group using independently differentiated hiPSC-CMs; mean ± SD; $p$ value by one sample $t$-test). (J) Superoxide anion content expressed as a percentage of empty AC16 cells compared to the 100% of Mel AC16 cells. (technical $n = 3$ per condition, biological $n = 4$ per group; mean ± SD; $p$ value by one sample $t$-test). (K) Modulation of the electron flux in mitochondria isolated from wild-type, Mel null and Mel over hearts, estimated as reduction rate of cytochrome C between complexes I and III. ($n = 5$ per group; mean ± SD; $p$ value by unpaired $t$-test). (L), Level of ATP in mitochondria isolated from wild-type, Mel null and Mel over hearts, detected by luciferin-luciferase assay. ($n = 5$ per group; mean ± SD; $p$ value by unpaired $t$-test). (M) Comparison of electron flux modulation and ATP levels between subsarcolemmal (SS) and intermyofibrillar (IMF) mitochondria isolated from Mel null, Mel over, and wild-type hearts. ($n = 3$ per group; mean ± SD; $p$ value by unpaired $t$-test). ROS reactive oxygen species, DCFDA-AM 5-(and-6)-chloromethyl-2′,7′-dichlorodihydro-fluorescein diacetate-acetoxymethyl ester, SS subsarcolemmal mitochondria, IMF intermyofibrillar mitochondria, TMZ trimetazidine, HFD high-fat diet, cit cytochrome C, EF electron flux. Source data are available online for this figure.

trifunctional protein (α-MTP and β-MTP) (Fig. EV1A,B). Melusin, as MTP subunits, was found as a component of the mitochondrial matrix in fractionation assays (Fig. EV1C,D), present both in subsarcolemmal and intermyofibrillar mitochondria (Fig. EV1E).

This interaction suggested a role for Melusin in the assembly of MTP multimeric complexes. Gel filtration experiments on mouse cardiac extracts showed Melusin in the same fractions containing both MTP α and β subunits, but did not show any changes in the multimeric organization of the MTP holoenzyme among Melusin-null (Brancaccio et al, 2003), Melusin overexpressing (De Acetis et al, 2005) and wild-type hearts (Fig. EV1F). Thus, Melusin is not required for MTP assembly.

We considered that Melusin might interact with the MTP to promote its catalytic activity and, ultimately, cardiac FAO. In contrast to this hypothesis, α-MTP activity was significantly higher in Melusin-null hearts (Fig. 1A) and lower in Melusin over-expressing hearts (Fig. EV2A) than in wild-type controls, indicating that Melusin unexpectedly limits FAO. To consolidate this finding, we evaluated the degree of palmitate oxidation in cardiac isolated mitochondria from mice either lacking or over-expressing Melusin. Palmitate oxidation capacity in Melusin-null mitochondria was ~35% higher than in wild-type controls (Fig. 1B), supporting our evidence that Melusin inhibits FAO and that the loss of Melusin leads to increased LCFA β-oxidation. Moreover, in mitochondria from hearts over-expressing ~20-fold higher Melusin (De Acetis et al, 2005), palmitate oxidation capacity was found ~55% lower than wild-type controls (Fig. 1B). The increased palmitate oxidation capacity detected in Melusin-null mitochondria was reversed by treatment with the MTP inhibitor trimetazidine (TMZ) (Fig. 1C,D), thus indicating that TMZ acts as an analog of Melusin on MTP.

Melusin levels did not affect the expression of MTP α and β subunits, as assessed by both qRT-PCR (Fig. EV2B) and western blot analyses (Fig. EV2C). The amount of the main lipid families (Fig. 1E; Table EV1) and the level of malonyl-CoA (Fig. EV2D), an important regulator of mitochondrial uptake of fatty acids, were comparable in the hearts of different genotypes, ruling out the possibility of a modification in lipid composition due to uncoupling between LCFA uptake and utilization (Yamamoto and Sano, 2022). We did not detect any difference in key molecules involved in glucose metabolism, such as Glucose transporter 1 and 4 (Glut1 and Glut4) mRNAs (Fig. EV2E) or pyruvate kinase and phosphorylated pyruvate dehydrogenase protein levels (Fig. EV2F). The preserved NADH/NAD$^+$ ratio also excluded an imbalance in the fueling of the mitochondrial respiratory chain (Fig. EV2G). Overall, these results indicate that Melusin affects FAO directly through its binding to MTP.

To assess the relevance of this finding in human cells, we generated Melusin-null and over-expressing human induced pluripotent stem cell-derived cardiomyocytes (hiPSC-CMs) (Figs. 1F and 2H–L). A proximity ligation assay (PLA) confirmed the interaction of Melusin and MTP also in these cells. (Fig. 1G). We determined a ~4.9-fold increase in palmitate β-oxidation capacity and a ~1.5-fold decrease in glucose flux through tricarboxylic acid (TCA) cycle in hiPSC-CMs compared to hiPSCs (Fig. 1H,I), in line with the expected change in metabolism between pluripotent and differentiated cells.

Melusin-null hiPSC-CMs boosted palmitate β-oxidation capacity by ~50% compared to wild-type controls, and this was matched by a ~3-fold reduction in glucose flux through the TCA cycle (Fig. 1H,I), possibly determined by the Randle cycle regulatory mechanism. These findings were observed in two separate Melusin-null hiPSC clones (Mel Null cl1 and cl2), ruling out the possibility of artefactual differences due to clonal variability. We further demonstrated that the higher palmitate oxidative capacity of Melusin-null cells reflects a real increase in palmitate utilization, by detecting a higher oxygen consumption rate in Melusin-null hiPSC-CMs than wild-type hiPSC-CMs when cultured in a media added with palmitate (Fig. 1J).

Melusin overexpression in hiPSC-CMs, by means of AAV vectors serotypes 6 and 9, led to dose-dependent increase in glucose processing

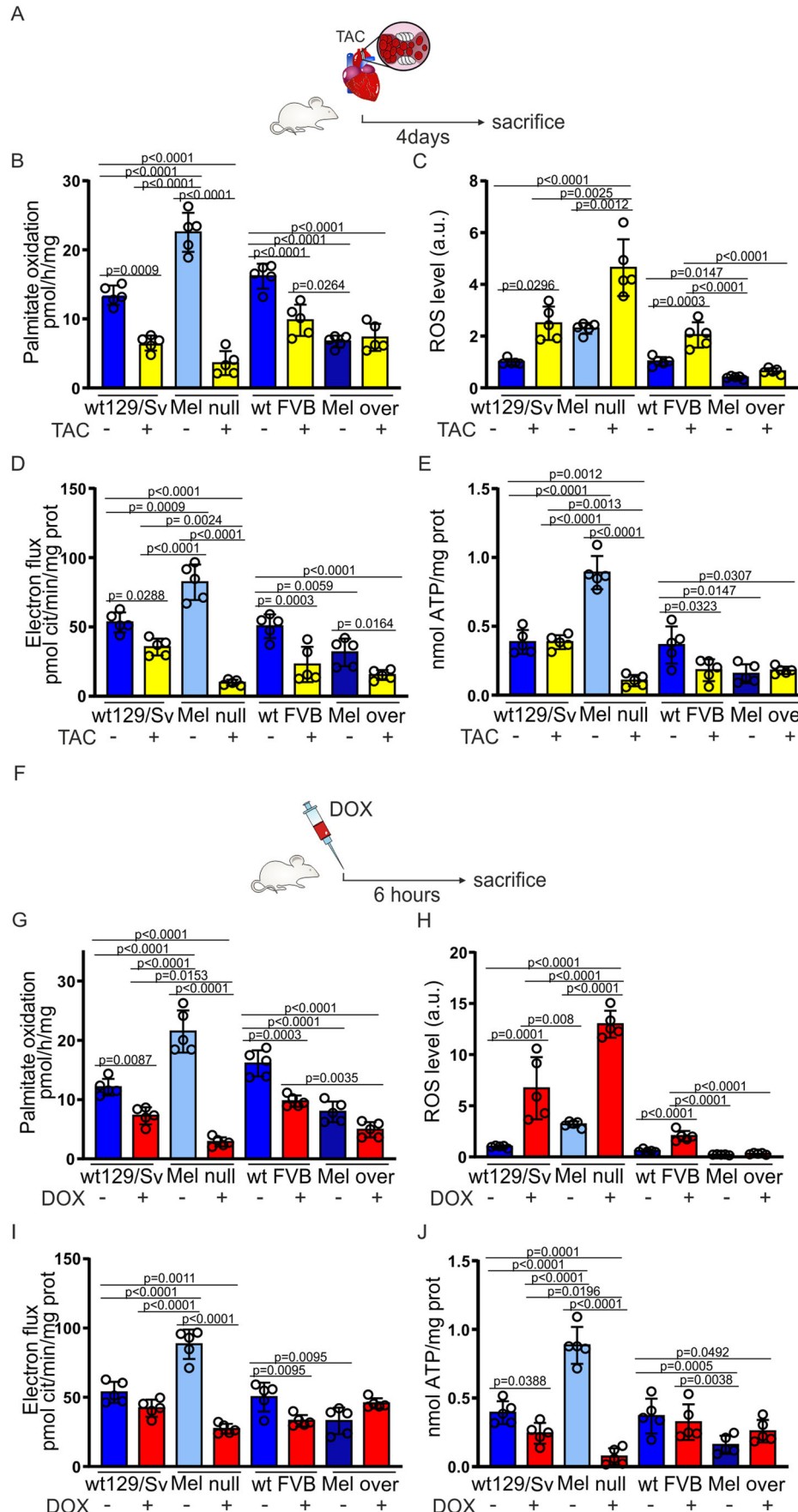

◄ **Figure 3. Melusin protects the heart from ROS production and mitochondrial dysfunction during stress.**

(A) TAC treatment protocol: Mel null, Mel over, and wild-type mice were anesthetized and subjected to transverse aortic banding (TAC +) to induce pressure overload in the heart. Control mice were subjected to sham surgery but without aortic banding (TAC −). After 4 days, mice were sacrificed to collect the hearts. (B) Modulation of palmitate oxidation in cardiac isolated mitochondria from Mel null, Mel over, and wild-type mice treated as described in (A). ($n = 5$ per group; mean ± SD; $p$ value by two-way ANOVA with Tukey correction (performed separately for the two mice strains)). (C) ROS content in cardiac isolated mitochondria from Mel null, Mel over, and wild-type mice treated as described in (A). ($n = 5$ per group; mean ± SD; $p$ value by two-way ANOVA with Tukey correction (performed separately for the two mice strains); Data normalized on wt129/Sv TAC - media set to 1). (D) Modulation of the electron flux, estimated as reduction rate of cytochrome C between complexes I and III, in mitochondria isolated from hearts of wild-type, Mel null and Mel over mice treated as described in (A). ($n = 5$ per group; mean ± SD; $p$ value by two-way ANOVA with Tukey correction (performed separately for the two mice strains)). (E) Level of ATP, detected by luciferin-luciferase assay, in mitochondria isolated from hearts of wild-type, Mel null and Mel over mice treated as described in (A). ($n = 5$ for each condition; mean ± SD; $p$ value by two-way ANOVA with Tukey correction (performed separately for the two mice strains)). (F) Doxorubicin single dose protocol in mice: Mel null, Mel over, and wild-type mice were injected intraperitoneally with a single dose of doxorubicin 4 mg/kg (DOX +) or saline (DOX −) as control. Mice were sacrificed 6 h later to collect the hearts. (G) Modulation of palmitate oxidation evaluated as radioactive metabolites of [1−14C]-palmitate generated in cardiac isolated mitochondria from Mel null, Mel over and wild-type mice treated as described in (F). ($n = 5$ per group; mean ± SD; $p$ value by two-way ANOVA with Tukey correction (performed separately for the two mice strains)). (H) ROS content was measured as DCFDA-AM fluorescence in cardiac isolated mitochondria from Mel null, Mel over, and wild-type mice treated as described in (F). ($n = 5$ per group; mean ± SD; $p$ value by two-way ANOVA with Tukey correction (performed separately for the two mice strains); Data normalized on wt129/Sv DOX - media = 1). (I) Modulation of the electron flux, estimated as reduction rate of cytochrome C between complexes I and III, in mitochondria isolated from hearts of wild-type, Mel null and Mel over mice treated as described in (F). ($n = 5$ per group; mean ± SD; $p$ value by two-way ANOVA with Tukey correction (performed separately for the two mice strains)). (J) Level of ATP, detected by luciferin-luciferase assay, in mitochondria isolated from hearts of wild-type, Mel null and Mel over mice treated as described in (F). ($n = 5$ for each condition; mean ± SD; $p$ value by two-way ANOVA with Tukey correction (performed separately for the two mice strains)). DCFDA-AM 5-(and-6)-chloromethyl-2′,7′-dichlorodihydro-fluorescein diacetate-acetoxymethyl ester, TAC transverse aortic constriction, cit cytochrome C, DOX doxorubicin; Source data are available online for this figure.

by the TCA cycle and decrease in palmitate oxidative capacity (Fig. 1H,I). Overall, these experiments indicate that the MTP inhibitory function of Melusin in cardiomyocytes is conserved between mouse and humans.

To further probe the activity of Melusin in other contexts, we also forced its expression in the human fibro-cardiac cell line AC16, a cell line that normally do not express this protein (Fig. 1K). PLA experiments demonstrated Melusin interaction with MTP also in these cells (Fig. EV2M). We found that Melusin expression in the AC16 cell line reduces the palmitate oxidation capacity and increases the glucose flux through the TCA cycle (Fig. 1L). In addition, the expression of Melusin decreased the real-time oxygen consumption rate when AC16 were cultured in media containing palmitate (Fig. 1M). Overall, these findings indicate that Melusin acts as a general inhibitor of MTP, even when expressed in non-contractile cells.

## Melusin inhibits ROS generation in the myocardium

Increased levels of FAO induce the production of more ROS by direct and indirect mechanisms (Battault et al, 2020; Isei et al, 2022; Mylonas et al, 2023; Speijer et al, 2014; St-Pierre et al, 2002), thus, in line with Melusin ability to inhibit FAO, we detected higher levels of ROS in Melusin-null hearts and lower levels in Melusin over-expressing hearts than in the respective controls (Fig. 2A). Specifically, ROS were 4-times higher in mitochondria from Melusin-null hearts than wild-type controls and almost three times lower in mitochondria from Melusin over-expressing cardiac muscles than their matched wild-type organs (Fig. 2A). Similar results were obtained by comparing the amount of ROS in subsarcolemmal and intermyofibrillar mitochondria from hearts of different genotypes (Fig. 2B). This regulation did not depend on differences in lipid peroxidation (Fig. EV3A), on the NADPH oxidative state (Fig. EV3B) and the level of NADPH oxidase 4 (Fig. EV3C), neither on relevant alterations in the expression of key antioxidant enzymes (Fig. EV3D). Moreover, treatment of Melusin-null mice with the MTP inhibitor TMZ restored ROS amounts to the wild-type levels (Figs. 1C and 2C), thus confirming that Melusin limits ROS generation by tuning MTP activity and FAO.

In further agreement, we found that a high-fat diet, boosting FAO, leads to a ~1.6-fold increase in palmitate oxidative capacity (Fig. 2D) and ~4-fold elevation in ROS production (Fig. 2E) in murine heart mitochondria. However, this induction of FAO and ROS generation did not occur when Melusin is over-expressed in the heart (Fig. 2D,E).

Next, we tested whether Melusin-mediated FAO repression, and the consequent reduction of ROS, could be confirmed in human cardiomyocytes. In line with murine models, ROS were ~2-fold higher in Melusin-null hiPSC-CMs (Fig. 2F), and, conversely, ~3-fold lower in Melusin over-expressing hiPSC-CMs (Fig. 2G) than controls. Results in hiPSC-CMs were also confirmed in AC16 cells expressing Melusin where ROS levels were reduced by ~3.7-fold (Fig. 2H). We detected an analogous modulation by analyzing specifically the level of superoxide anion through MitoSOX fluorescence in living wild-type and Melusin-null hiPSC-CMs (Fig. 2I) and in AC16 expressing or not Melusin (Fig. 2J). All these data support the idea that Melusin regulates the generation of mitochondrial ROS.

A functional analysis of mitochondria isolated from mouse hearts showed that the electron flux through the respiratory chain correlates with the rate of FAO, being >60% higher in Melusin-null mice (Fig. 2K). ATP levels are similarly anti-correlated with Melusin expression (Fig. 2L). These differences were detected both in subsarcolemmal and intermyofibrillar mitochondria (Fig. 2M). This evidence was not explained by the different abundance of mitochondria, as indicated by comparable amounts of the mitochondrial proteins Vdac1 and Tom20 in cardiac protein lysates (Fig. EV2E). Similarly, the expression of the five mitochondrial respiratory chain complexes was equivalent in all genotypes (Fig. EV3E). Of note, no significant differences were detected in measuring the ATP content in the entire heart tissues, suggesting that cytoplasmic processes may mostly compensate for the differential ATP production in mitochondria (Fig. EV3F), in line with the healthy phenotype of Melusin overexpressing mice.

The physiologic restriction of FAO mediated by Melusin did not block the increase in lipid metabolism associated with the higher

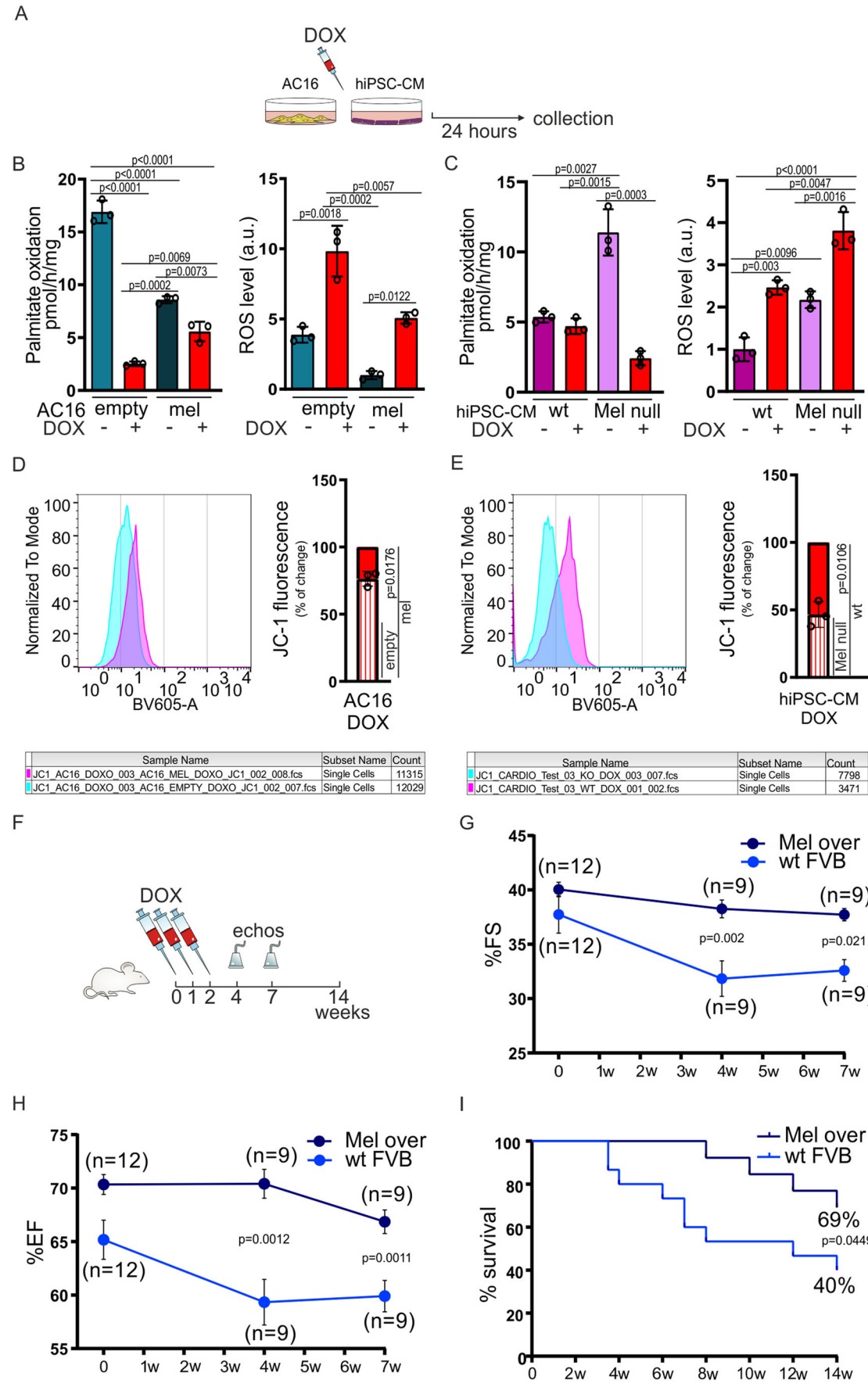

◄

**Figure 4. Melusin protects the heart from doxorubicin cardiotoxicity restraining ROS production and mitochondrial dysfunction, preserving the cardiac function, and sustaining the overall survival.**

(A) Doxorubicin protocol in cells: AC16 cells empty e mel and hiPSC-CMs wild-type and null for Melusin were treated with 250 nM doxorubicin (DOX +) or normal media (DOX −) for 24 h before cell collection for analyses. (B) Modulation of palmitate oxidation (left) and ROS content (right) in mitochondria from AC16 cells empty and mel, treated as described in (A). ($n = 3$ mitochondrial preparations from independent experiments; mean ± SD; p value by two-way ANOVA with Tukey correction; ROS data normalized on AC16 mel DOX - media = 1). (C) Modulation of palmitate oxidation (left) and ROS content (right) in mitochondria from hiPSC-CMs wild-type and null for Melusin expression, treated as described in (A). ($n = 3$ mitochondrial preparations per group from hiPSC-CMs independently differentiated ; mean ± SD; p value by two-way ANOVA with Tukey correction; ROS data normalized on hiPSC-CMs wt DOX - media = 1). (D) Percentage of mitochondria with normal membrane potential, evaluated as red fluorescence of JC-1 probe by FACS analysis, in AC16 cells empty and expressing Melusin, treated as described in (A). A representative histogram is reported on the left, and the percentage of change graph is reported on the right. Empty AC16 cells are reported as a percentage of Mel AC16 settled as 100%. (technical $n = 2/3$ per condition, biological $n = 3$ per group; mean ± SD; p value by one sample t-test). (E) Percentage of mitochondria with normal membrane potential in hiPSC-CMs wild-type and null for Melusin, treated as described in (A). A representative histogram is reported on the left, and the percentage of change graph is reported on the right. Mel null hiPSC-CMs are reported as a percentage of wild-type hiPSC-CMs settled as 100%. (technical $n = 2/3$ per condition, biological $n = 3$ per group; mean ± SD; p value by one sample t-test). (F) Doxorubicin multiple dose protocol in mice: Mel over and wild-type mice were injected intraperitoneally with three doses of doxorubicin 4 mg/kg at $t = 0$, $t = 1$ week, and $t = 2$ weeks. Echocardiographic analyses were performed at 4 and 7 weeks. Mice were followed for 14 weeks for survival. (G) Percentage of fractional shortening detected by echo analysis at 4 and 7 weeks in Mel over and wild-type mice treated as described in (F). (nt0 = 12 Mel over, 12 wt FVB; nt4w = 9 Mel over, 9 wt FVB; nt7w = 9 Mel over, 9 wt FVB; mean ± SEM; p value by two-way ANOVA with Sidak correction). (H) Percentage of ejection fraction detected by echo analysis at 4 and 7 weeks in Mel over and wild-type mice treated as described in (F). (nt0 = 12 Mel over, 12 wt FVB; nt4w = 9 Mel over, 9 wt; nt7w = 9 Mel over, 9 wt FVB; mean ± SEM; p value by two-way ANOVA with Sidak correction). (I) Percentage of survival until 14 weeks in Mel over and wild-type mice treated as described in (F). ($n = 13$ Mel over, $n = 15$ wt FVB; analysis by Mantel–Cox test). DOX doxorubicin, DCFDA-AM 5-(and-6)-chloromethyl-2',7'-dichlorodihydro-fluorescein diacetate-acetoxymethyl ester, cit cytochrome C, %FS, percentage of fractional shortening, %EF percentage of ejection fraction. Source data are available online for this figure.

energetic demand induced by β-adrenergic stimulation (Banke et al, 2010). Indeed, simulating intense exercise by treating the isolated perfused heart for 10 min with a mild concentration of isoproterenol (Fig. EV3G), we detected an increase in FAO in wild-type hearts (Fig. EV3H), while no further increase was observed in Melusin-null hearts (Fig. EV3H). To note, the 20-fold over-expression of Melusin blocks the increase in FAO induced by mild β-adrenergic stimulation (Fig. EV3H). This suggests that, in the absence of Melusin, cardiomyocytes metabolize lipids to their highest potential, while physiological levels of Melusin tune this process, restraining ROS production. The amount of ROS observed in isoproterenol-treated hearts of different genotypes (Fig. EV3I) likely depends on the basal level of ROS in these hearts added to ROS generated by isoproterenol per se (Dey et al, 2018).

Overall, these data demonstrate that by inhibiting MTP-dependent FAO, Melusin limits the over-production of ROS and regulates mitochondrial function.

## Melusin protects the heart from ROS production and mitochondrial dysfunction in pressure overload

We previously highlighted how Melusin overexpression protects the heart from pressure overload induced by transverse aortic constriction (TAC) (De Acetis et al, 2005), while Melusin-null mice develop left ventricle dilation and heart failure in 4 weeks (Brancaccio et al, 2003). To evaluate whether the ability of Melusin to regulate metabolism plays a role in TAC protection, we subjected Melusin-null and over-expressing mice, together with wild-type controls, to 4-day TAC or sham surgery (Fig. 3A). This early time point was chosen since it precedes the TAC-induced cardiac remodeling (Brancaccio et al, 2003; De Acetis et al, 2005). At 4 days after TAC, FAO mildly (~37%) decreased in wild-type mice, in line with the metabolic rewiring occurring in the heart during stress. Unexpectedly, FAO appeared potently reduced in Melusin-null hearts (Fig. 3B), apparently in contrast with the role of Melusin in limiting MTP activity. We posited that this result might be due to a mitochondrial dysfunction triggered by the ROS increase induced by ROS generated by pressure overload per se (Suetomi et al, 2018; Wang et al, 2018), added to the basal ROS

level derived from FAO. Indeed, we detected an abnormally high ROS production (Fig. 3C). In contrast, FAO rate and ROS levels in Melusin over-expressing mice did not vary in response to pressure overload (Fig. 3B,C). ROS levels did not depend on differences in the level of NADPH oxidase 4 among genotypes (Fig. EV4A).

In line with these observations, the mitochondrial electron flux, reduced by ~2.7 and ~2.4-fold, respectively in wild-type and Melusin over-expressing hearts (Fig. 3D), is ~8.6-fold reduced in Melusin-null hearts (Fig. 3D) in response to TAC. We further detected no changes in mitochondrial ATP in all genotypes, except for Melusin-null hearts where a ~7.5-fold drop was observed (Fig. 3E). These alterations occurred in the absence of significant variations in MTP expression (Fig. EV4A) and differences in mitochondrial content (Fig. EV4B, C) or structure (Fig. EV4D,E). In line, we measured the ATP in the cardiac total homogenate of all the mouse genotypes and found a 3-fold drop in total ATP content in Melusin-null hearts subjected to TAC (Fig. EV4F).

Overall, these results show that, at early stages during pressure overload, Melusin protects the heart from the ROS over-production and the consequent drop of the mitochondrial function.

## Melusin prevents ROS boost and mitochondrial dysfunction during cardiotoxicity induced by doxorubicin

The rise of cardiac ROS production is thought to occur in response to doxorubicin, a commonly used and effective antineoplastic drug with cardiotoxic side effects (Berthiaume and Wallace, 2007; Brancaccio et al, 2020). We thus tested whether Melusin, by controlling FAO and ROS production, could affect the cardiac response to doxorubicin.

First, we analyzed the effect of doxorubicin on FAO and ROS production in acute, 6 h after doxorubicin treatment (Fig. 3F). At this time point, FAO decreased in wild-type and Melusin overexpressing mice (~40% and 50%, respectively) (Fig. 3G), while, despite its higher starting point, drastically dropped (~88%) in Melusin-null hearts (Fig. 3G), similarly to the TAC model.

On the other hand, total ROS content was significantly increased by the doxorubicin treatment itself both in wild-type and Melusin-

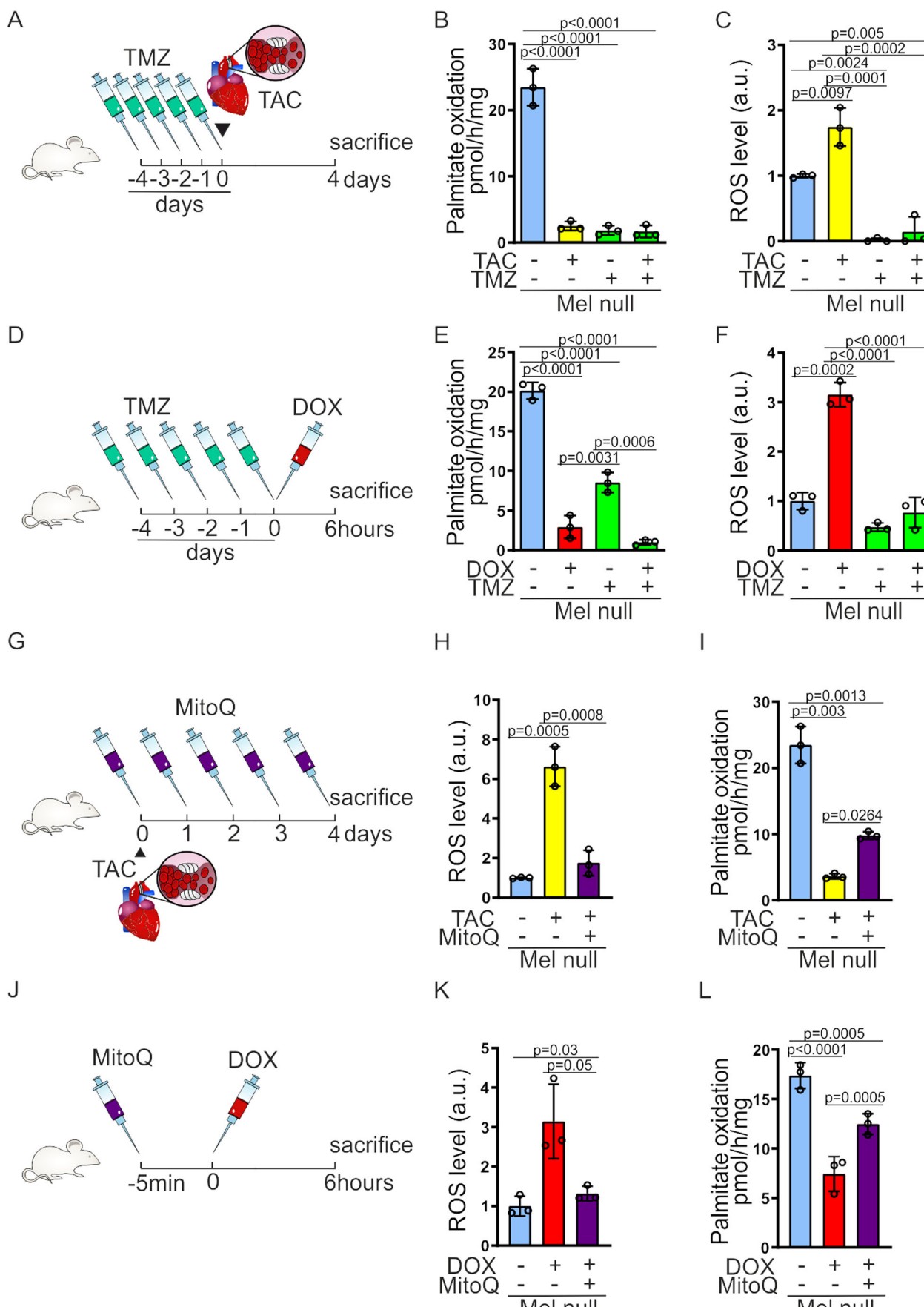

Figure 5. The level of FAO and ROS in steady state impacts on the cardiac response to stress.

(A) Trimetazidine and TAC treatment protocol: Mel null mice were treated orally for 5 days with 30 mg/kg TMZ (TMZ +) or saline (TMZ −) as control. At day 0 mice were subjected to TAC (TAC +) or sham (TAC −) surgery. After 4 days, mice were sacrificed to collect the hearts. (B) Modulation of palmitate oxidation estimated as radioactive metabolites of [1−14C]-palmitate generated in cardiac isolated mitochondria from Mel null mice treated as described in (A). (n = 3 per group; mean ± SD; p value by two-way ANOVA with Bonferroni correction). (C) ROS content evaluated as DCFDA-AM fluorescence in cardiac isolated mitochondria from Mel null mice treated as described in (A). (n = 3 per group; mean ± SD; p value by two-way ANOVA with Bonferroni correction; Data normalized on Mel null TAC-/TMZ- media = 1). (D) Trimetazidine and doxorubicin treatment protocol: Mel null mice were treated orally for 5 days with 30 mg/kg TMZ (TMZ +) or saline (TMZ −) as control. At day 0 mice were injected intraperitoneally with a single dose of doxorubicin 4 mg/kg (DOX +) or saline (DOX −) as control. Mice were sacrificed 6 h later to collect the hearts. (E) Modulation of palmitate oxidation in cardiac isolated mitochondria from Mel null mice treated as described in (D). (n = 3 per group; mean ± SD; p value by two-way ANOVA with Bonferroni correction). (F) ROS content in cardiac isolated mitochondria from Mel null mice treated as described in (D). (n = 3 per group; mean ± SD; p value by two-way ANOVA with Bonferroni correction; Data normalized on Mel null DOX-/TMZ- media = 1). (G) MitoQ and TAC treatment protocol: at day 0 Mel null mice were subjected to TAC (TAC +) or sham (TAC −) surgery. Starting from day 0, mice were treated daily with an oral administration of 2 μmol MitoQ (MitoQ +) or the same volume of saline (MitoQ −) as control. Mice were finally sacrificed on day 4 to collect the hearts. (H) ROS content in cardiac isolated mitochondria from Mel null mice treated as described in (G). (n = 3 per group; mean ± SD; p value by two-way ANOVA with Bonferroni correction; Data normalized on Mel null TAC-/MitoQ- media = 1). (I) Modulation of palmitate oxidation in cardiac isolated mitochondria from Mel null mice treated as described in (G). (n = 3 per group; mean ± SD; p value by two-way ANOVA with Bonferroni correction). (J) MitoQ and doxorubicin treatment protocol: Mel null mice were administered orally with 2 μmol MitoQ (MitoQ +), or saline as control (MitoQ −), 5 min before being injected with a single dose of doxorubicin 4 mg/kg (DOX +) or saline (DOX −) as control. Mice were sacrificed 6 h later to collect the hearts. (K) ROS content in cardiac isolated mitochondria from Mel null mice treated as described in (J). (n = 3 per group; mean ± SD; p value by two-way ANOVA with Bonferroni correction; Data normalized on Mel null DOX-/MitoQ- media = 1). (L) Modulation of palmitate oxidation in cardiac isolated mitochondria from Mel null mice treated as described in (J). (n = 3 per group; mean ± SD; p value by two-way ANOVA with Bonferroni correction). TMZ trimetazidine, TAC transverse aortic constriction, DCFDA-AM 5-(and-6)-chloromethyl-2',7'-dichlorodihydro-fluorescein diacetate-acetoxymethyl ester, DOX doxorubicin, MitoQ mitoquinone mesylate. Source data are available online for this figure.

null mice proportionally to their respective basal states, with Melusin-null mice showing consistently higher ROS than wild-type controls (Fig. 3H). Conversely, Melusin over-expressing hearts showed lower ROS levels in basal conditions that did not increase after doxorubicin treatment (Fig. 3H). The high level of ROS in doxorubicin-treated Melusin-null mice can explain the mild significant increase in the level of NADPH oxidase 4 in these mice, while no differences were detected in treated Melusin over-expressing mice compared to wild-type mice (Fig. EV5A).

We further analyzed the mitochondrial function after 6 h from doxorubicin administration, detecting alterations like those observed in pressure overload. Indeed, in Melusin-null mice the electron flux rate was reduced by ~71% after doxorubicin treatment, while no changes were detected in wild-type and Melusin over-expressing mice (Fig. 3I). Mitochondrial ATP levels showed a ~96% drop in Melusin-null mice, in accordance with the decrease in FAO, while no significant differences were detected in wild-type and over-expressing mice (Fig. 3J). Similarly to TAC, this finding did not correlate with detectable changes in the expression of MTP (Fig. EV5A), mitochondrial content (Fig. EV5B,C) or structure (Fig. EV5D,E), and mitochondrial integrity (Fig EV5F). By measuring the total amount of ATP in heart tissues, we detected a decrease of ~60% in Melusin-null mice in response to doxorubicin treatment (Fig. EV5G).

We then treated with doxorubicin AC16 cells wild-type and expressing Melusin and hiPSC-CMs wild-type and knockout for Melusin (Fig. 4A), showing that Melusin protects cells from the drop of FAO and the rise of ROS (Fig. 4B,C), in line with data obtained in mice. In particular, Melusin absence in AC16 and hiPSC-CMs correlates with a lower number of mitochondria with normal membrane potential after doxorubicin treatment (Fig. 4D,E), indicating that Melusin protects cardiomyocytes from mitochondrial dysfunction.

Finally, given the protective effect of Melusin overexpression after an acute doxorubicin challenge, we studied whether increased Melusin abundance could mitigate cardiotoxicity induced by a repeated doxorubicin treatment (Fig. 4F). We found that Melusin overexpression leads to preserved fractional shortening (Fig. 4G)

and ejection fraction (Fig. 4H), as well as reduced cumulative mortality (Fig. 4I).

These results demonstrate that in response to doxorubicin treatment, similarly to what we observed after pressure overload, the modulatory role of Melusin on FAO preserves the heart from an excessive ROS generation and from mitochondrial dysfunction.

## The rate of fatty acid oxidation in physiologic conditions impacts on the cardiac response to stress

The sharp decline in FAO observed in Melusin-null mitochondria after challenges, like TAC and doxorubicin, suggested that the higher basal levels of ROS present in these hearts in steady state, boosted by the ROS induced by the stress itself, induce mitochondrial dysfunction and the consequent drop of FAO. Therefore, we tested if keeping FAO low before stress, reduces the ROS levels generated after the cardiac challenge. We treated Melusin-null mice with the MTP inhibitor trimetazidine (TMZ) before either TAC (Fig. 5A,B) or doxorubicin administration (Fig. 5D,E) and we revealed that, after both challenges, pre-treatment with TMZ potently reduced ROS in Melusin-null hearts (Fig. 5C, F). Next, we tested if reducing ROS with the MitoQ scavenger concomitantly to the challenge could mitigate the mitochondrial dysfunction and the observed drop of FAO (Fig. 5G,J). In line with ROS being critical in this event, MitoQ blocked the raise of ROS levels (Fig. 5H,K) and dampen the severe decline of FAO seen in Melusin-null hearts (Fig. 5I,L).

These results indicate that inhibiting MTP and lowering FAO reduce the burden of ROS accumulation during stress-limiting mitochondrial dysfunction.

## Limiting fatty acid oxidation in concomitance to stress sustains cardiac function

The reduction of FAO timely during stress might improve overall cardiac resilience. To test this hypothesis, we treated wild-type mice with TMZ concomitantly to pressure overload as well as doxorubicin administration. TMZ dissolved in drinking water

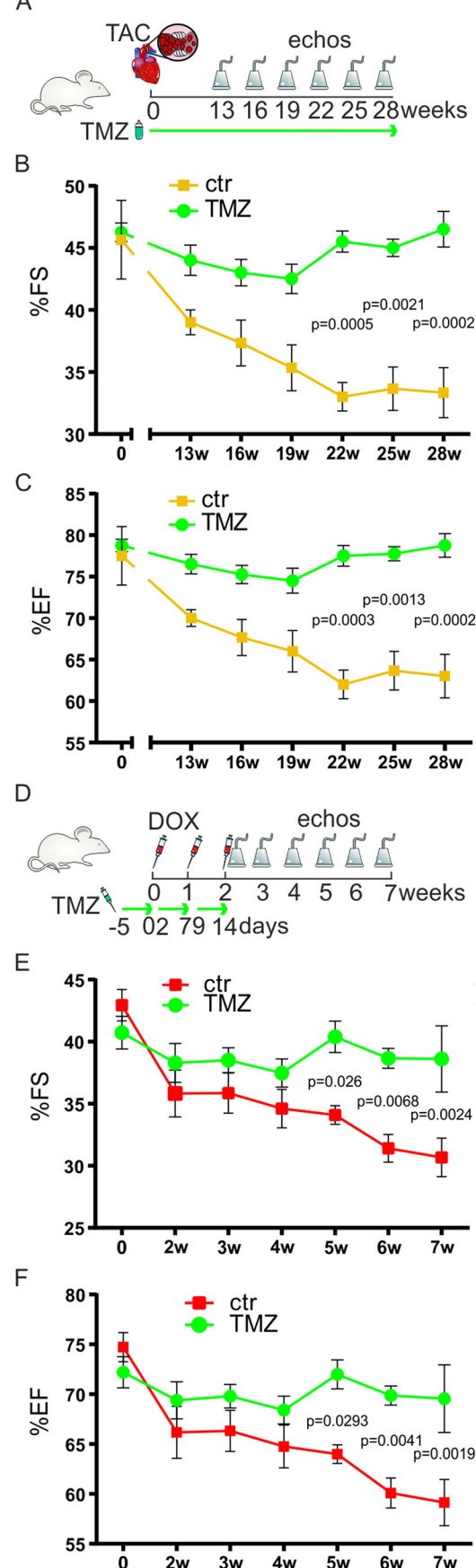

**Figure 6. Tuning FAO before or concomitantly to cardiac challenge sustains myocardial contractility.**

(A) Trimetazidine and long TAC protocol: wild-type mice were subjected to TAC surgery and treated with 0.7 g/L TMZ dissolved in drinking water, or normal water as control. Mice were monitored by echo analyses every 3 weeks, from week 13th until week 28th. (B) Percentage of fractional shortening detected by echo analysis in wild-type mice treated as described in (A). ($n = 3$ ctr, $n = 4$ TMZ; mean ± SEM; $p$ value by two-way ANOVA with Sidak correction). (C) Percentage of ejection fraction detected by echo analysis in wild-type mice treated as described in (A). ($n = 3$ ctr, $n = 4$ TMZ; mean ± SEM; $p$ value by two-way ANOVA with Sidak correction). (D) Trimetazidine and doxorubicin multiple dose protocol: wild-type mice were administered orally for 5 days with 30 mg/kg TMZ, or saline as control, before each treatment with doxorubicin. Mice received three doses of 4 mg/kg doxorubicin at $t = 0$, $t = 1$ week, and $t = 2$ weeks. Mice were monitored by echo analyses weekly, from week 2 to week 7. (E) Percentage of fractional shortening detected by echo analysis in wild-type mice treated as described in (D). ($n = 5$ ctr, $n = 5$ TMZ; mean ± SEM; $p$ value by one-way ANOVA with Sidak correction). (F) Percentage of ejection fraction detected by echo analysis in wild-type mice treated as described in (D). ($n = 5$ ctr, $n = 5$ TMZ; mean ± SEM; $p$ value by one-way ANOVA with Bonferroni correction). TMZ trimetazidine, TAC transverse aortic constriction, DOX doxorubicin, %FS percentage of fractional shortening, %EF percentage of ejection fraction. Source data are available online for this figure.

(Fig. 6A) preserved fractional shortening and ejection fraction for 28 weeks after TAC (Fig. 6B,C). Similarly, the oral administration of TMZ concomitantly with three doses of doxorubicin (Fig. 6D) protected hearts from the decrease in fractional shortening and ejection fraction seen in vehicle-treated mice (Fig. 6E,F). In conclusion, these experiments demonstrate that approaches aimed at mitigating lipid oxidation concomitantly to challenge improves the ability of the heart to cope with stress (Fig. 7).

## Discussion

In this study, we revealed an unexpected mechanism of metabolic regulation in the heart that, settled in physiologic conditions, creates the basis for cardiac resilience to stressful events. This mechanism, conserved between mice and humans, is exerted by the cardioprotective chaperone Melusin through its interaction with the MTP, the key enzyme for long-chain fatty acid catabolism. We found that Melusin enters mitochondria, binds to MTP, and inhibits its activity, restraining the myocardial FAO rate.

FAO is the main energetic metabolism of the heart in a steady state, but it leads to the formation of ROS (Speijer et al, 2014; St-Pierre et al, 2002). ROS generation due to FAO may depend on different mechanisms. NADH and $FADH_2$ fuel mitochondrial Complex I and Complex II, respectively, and when FAO is high, $FADH_2$ increases electrons from Complex II, competing with the ones from Complex I for access to Complex III. This increases the probability of reverse electron transport within Complex I and electron leakage, causing ROS production (Murphy, 2009; Speijer et al, 2014). In addition, dehydrogenases involved in FAO reduce the electron transfer flavoprotein (ETF), which transfers electrons to the ETF-ubiquinone oxidoreductase, and this may cause electron leakage (Isei et al, 2022). Moreover, intermediates of fatty acid catabolism may inhibit the electron transport chain, favoring electron escape, and interfere with ROS scavenging (Seifert et al, 2010; St-Pierre et al, 2002). Furthermore, the preference for FAO

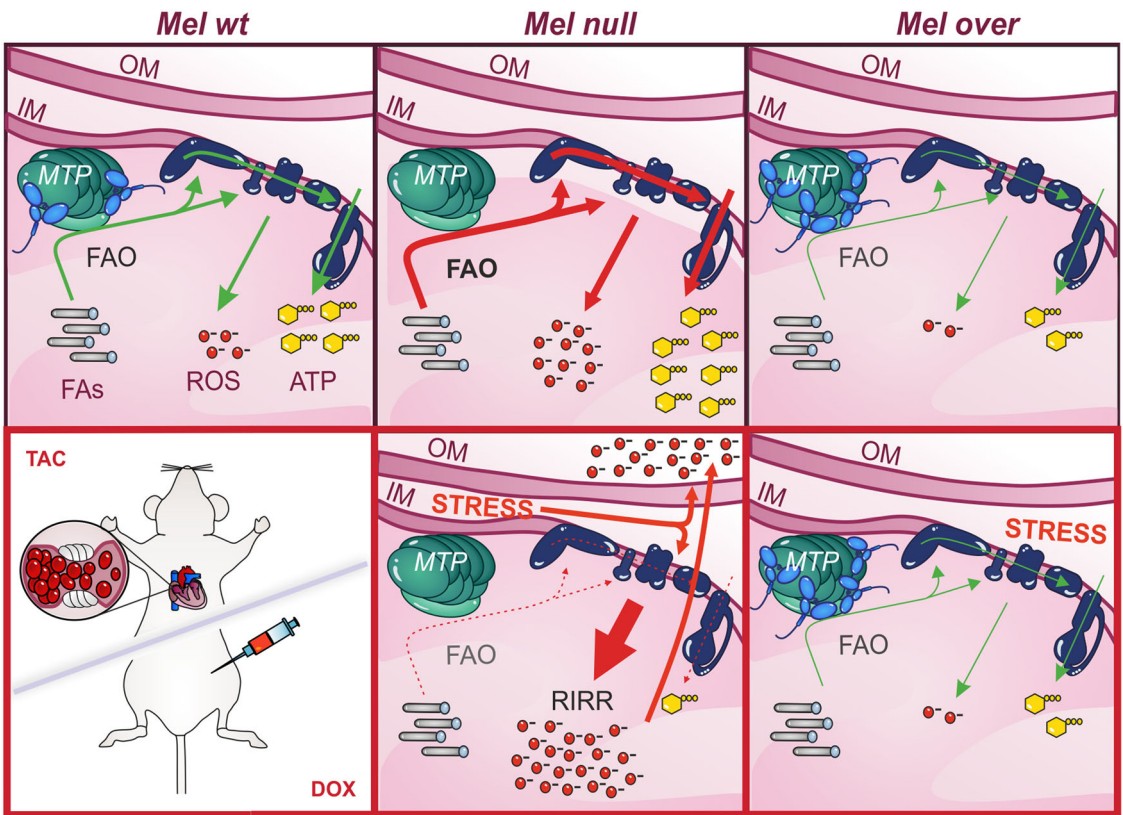

**Figure 7. Working model for Melusin promotion of cardiac resilience in mice.**

Top: in Mel wt mice, Melusin binds and inhibits the MTP, restraining the FAO rate. This tunes the electron flux through the respiratory chain, allows an adequate ATP synthesis, and limits the generation of ROS. In Mel null mice, the absence of Melusin causes an excessive FAO rate that, in turn, enhances the electron flux through the respiratory complexes, generates more ATP but increases the risk of electron leakage and ROS generation. In Mel over mice, more Melusin binds and inhibits the MTP, reducing the FAO rate. This limits the electron flux and the ATP production, still at adequate levels for cardiac function, and significantly decreases ROS level. Bottom: stress conditions, as doxorubicin treatment or aortic constriction, generate ROS per se that, added to the basal level induced by FAO, boost the generation of ROS from multiple sites. In Mel null mice, showing a high basal level of ROS, stressful stimuli activate the RIRR process and the generation of ROS from mitochondrial and cytosolic sources. This causes mitochondrial dysfunction with a consequent severe drop in FAO together with the mitochondrial electron flux and the ATP synthesis. Conversely, stress conditions fail in boosting the generation of ROS in Mel over mice, since their very low basal level of ROS. This protects mitochondria from damage and preserves their FAO level, electron flux, and energetic production. MTP mitochondrial trifunctional protein, OM outer membrane, IM inner membrane, FAO fatty acid oxidation, FAs fatty acids, TAC transverse aortic constriction, DOX doxorubicin.

may redirect glucose toward catabolic pathways different from mitochondrial oxidation, such as the polyol pathway and the hexosamine biosynthesis pathway, known to generate oxidative stress (Battault et al, 2020; Mylonas et al, 2023). In addition, the reduction of glycolysis decreases the production of serine, which has antioxidant effects (Mylonas et al, 2023).

ROS have harmful effects on the heart, damaging mitochondria, cell membranes, and cytoskeletal proteins, when they exceed a threshold that activates the auto-amplification process, known as RIRR (Tse et al, 2016; Tullio et al, 2013; Zorov et al, 2000).

Melusin, by binding the MTP in mitochondria, constitutively limits FAO and restrains the related ROS generation into a certain range. Indeed, in the absence of Melusin, FAO is dysregulated and ROS levels are high already in steady state (Fig. 7). How Melusin is regulated is still unclear. It is known that in the cytosol Melusin activates AKT and ERK pathways in response to mechanical signals, mediating cardiomyocytes survival and compensatory hypertrophy (Brancaccio et al, 2003; De Acetis et al, 2005; Penna et al, 2014; Sbroggio et al, 2011; Sorge and Brancaccio, 2016;

Unsold et al, 2014). We speculate that Melusin could shuttle from cytoplasm to mitochondria, or vice versa, in response to specific signals, as, for example, the availability of high-energy phosphates. Indeed, Melusin preferentially binds to the ADP-bound HSP90 (Gano and Simon, 2010; Hong et al, 2013), and this could represent an expedient to retain Melusin in the cytoplasm and burst MTP activity and ATP production when ADP/ATP ratio is high. Moreover, Melusin is provided with a C-terminal calcium-binding domain and two Cysteine- and Histidine-rich domains (CHORD domains) (Brancaccio et al, 1999). The C-terminal region can bind to calcium at low affinity but high capacity and this could modulate Melusin activity and location in response to calcium changes in the cytosol and in mitochondria. CHORD domains, whose folding requires coordination of a zinc ion by cysteines, can be reduced by ROS and, in turn, dictate conformational changes in Melusin that could modulate its localization and/or activity.

In this study, we defined the crucial role of Melusin in regulating ROS levels in the heart, already at steady state. This ability becomes crucial after challenges as different as pressure overload and

doxorubicin treatment, two conditions known to significantly increase the amount of ROS in cardiomyocytes (Berthiaume and Wallace, 2007; Brancaccio et al, 2020; Dai et al, 2011; Dey et al, 2018; Nickel et al, 2015; Okamura et al, 2019; Suetomi et al, 2018; Wang et al, 2018). When ROS from challenges, added to those generated in basal condition from FAO, reach a level that activates the RIRR vicious cycle (Dey et al, 2018; Leach et al, 2001; Tse et al, 2016; Tullio et al, 2013; Zorov et al, 2000) they cause mitochondrial dysfunction and compromise energy production (Aon et al, 2003; Dai et al, 2011; Dey et al, 2018; Tsutsui et al, 2011). Indeed, both pressure overload and doxorubicin treatment induce a higher ROS level in Melusin-null mice than in wild-type controls together with a drop in mitochondrial FAO and ATP production. This mitochondrial impairment depends on the excessive ROS production, since treatment with ROS scavengers partially rescues the ability of Melusin-null hearts to perform FAO after challenges. However, a translational use of antioxidant drugs to reduce ROS in the heart of patients showed low benefits in clinical trials (Heart Outcomes Prevention Evaluation Study et al, 2000; Heart Protection Study Collaborative, 2002) likely due to the fact that a finely tuned ROS generation is necessary to modulate crucial signaling events in the myocardium, such as calcium handling and excitation/contraction coupling (Burgoyne et al, 2012; Chen et al, 2014).

In stress conditions, the heart undergoes a shift from fatty acids toward glucose utilization (Akki et al, 2008; Kolwicz et al, 2013; Stanley et al, 2005; Zhong et al, 2013) and this may represent a physiological attempt to reduce the production of ROS when their level is increased by stress. Along the years, several approaches have been tried aimed at conditioning the heart toward glucose or fatty acid metabolism in a full-blown pathologic state to counteract the progression toward heart failure, obtaining contrasting results (Chiu et al, 2005; Choi et al, 2016; Jaswal et al, 2011; Kolwicz et al, 2013; Lionetti et al, 2011; Luptak et al, 2007; Yan et al, 2009; van Bilsen et al, 2009).

The findings of our study have to be seen in light of some limitations. While for hiPSC-CMs and AC16 cells, palmitate oxidation was analyzed also in living cells, for mouse hearts we only determined the maximal palmitate oxidation rate of isolated mitochondria from hearts. A metabolic tracing analysis determining the actual usage of metabolites in intact beating hearts will be an interesting subject of future study. We also relied on DCFDA-AM for measuring mitochondrial ROS: while this probe is widely used, the interpretation of results may be confounded by aspecific reactions besides superoxide. We performed specific staining for superoxide anions using MitoSOX and we observed a lower magnitude of differences than for DCFDA-AM, suggesting that other ROS species are involved in the process.

Overall, our findings demonstrate that Melusin and its drug-analogue TMZ sustain the cardiac resilience to stress by mitigating FAO rate already in steady state. This supports the idea that the reduction of myocardial FAO is beneficial when induced before or concomitantly to cardiac stress. Acting before stressful events is usually hard in clinics, but prevention is an established concept in cardiovascular protection, where, for example, drugs limiting blood pressure are commonly used. Thus, the possibility to limit FAO in case of risk of harmful events or in anticipation of therapies potentially impacting myocardial function could represent an innovative approach to potentiate cardiac resilience and prevent heart failure.

# Methods

### Reagents and tools table

| Reagent/resource | Reference or source | Identifier or catalog number |
| --- | --- | --- |
| **Experimental models** | | |
| FVB (*M. musculus*) | De Acetis et al, 2005 | |
| 129/Sv (*M. musculus*) | Brancaccio et al, 2003 | |
| WTC11 hiPSCs | Kreitzer et al, 2013 | |
| AC16 | Millipore | #SCC109 |
| **Recombinant DNA** | | |
| AAV6 and AAV9 | This work + Vitadello et al, 2020 | |
| pLVX lentivirus | This work | |
| pX459-V2.0_ITGB1BP2_-302 | This work | |
| pX459-V2.0_ITGB1BP2_+135 | This work | |
| pSpCas9 (BB)-2A-Puro (PX459) V2.0 | Addgene | #62988 |
| **Antibodies** | | |
| Mouse anti-Melusin | Sbroggiò et al, 2011 | |
| Mouse anti-vinculin | Sigma-Aldrich | SAB4200080 |
| Rabbit anti-HADHA | Abcam | ab203114 |
| Mouse anti-HADHB | Santa Cruz Biotechnology | sc-365907 |
| Rabbit anti-VDAC1 | Abcam | ab15895 |
| Mouse anti-actin | Santa Cruz Biotechnology | sc-1616-R |
| Rabbit anti-tom20 | Santa Cruz Biotechnology | sc-11415 |
| Mouse anti-total OXPHOS WB antibody cocktail | Abcam | ab110413 |
| Rabbit anti-pyruvate kinase PKM2 | Thermo Fisher Scientific | MA5-14890 |
| Rabbit anti-phospho-pyruvate dehydrogenase α1 (ser293) | Cell signaling | 31866S |
| Rabbit anti-pyruvate dehydrogenase | Cell signaling | 3205 |
| Rabbit anti-catalase | Abcam | ab16731 |
| Rabbit anti-glutathione peroxidases 1 | Thermo Fisher Scientific | #PA5-26323 |
| Mouse anti-glutathione peroxidases 4 | Santa Cruz Biotechnology | sc-166570 |
| Rabbit anti-peroxiredoxin 1 | Abcam | ab109498 |
| Rabbit anti-peroxiredoxin 3 | Abcam | ab222807 |
| Rabbit anti-superoxide dismutase 1 | Santa Cruz Biotechnology | sc-11407 |
| Mouse anti-superoxide dismutase 2 | Santa Cruz Biotechnology | sc-133134 |
| Mouse membrane integrity wb antibody cocktail | Abcam | ab110414 |

| Reagent/resource | Reference or source | Identifier or catalog number |
|---|---|---|
| Rabbit anti-Nox4 | Thermo Fisher Scientific | #14347-1-AP |
| Mouse anti-Myc tag | Thermo Fisher Scientific | MA1-21316 |
| Goat anti-rabbit IgG-peroxidase antibody | Sigma-Aldrich | #A6154 |
| Goat anti-mouse IgG-peroxidase antibody | Sigma-Aldrich | #A4416 |
| Goat anti-rabbit secondary antibody Alexa Fluor™ 633 | Thermo Fisher Scientific | A21070 |
| Goat anti-mouse secondary antibody Alexa Fluor™ 488 | Thermo Fisher Scientific | A11001 |
| Mouse Anti-Cardiac Troponin T Alexa Fluor 647 clone 13-11 | BD Pharmigen | 565744 |
| Mouse IgG1, κ Isotype Control Alexa Fluor 647 clone MOPC-31 C | BD Pharmigen | 566011 |
| **Oligonucleotides and other sequence-based reagents** | | |
| 18S | Thermo Fisher Scientific | 4319413E |
| PCR primers | This work | Methods section |
| **Chemicals, enzymes and other reagents** | | |
| Doxorubicin | Sigma-Aldrich | D1515 |
| Trimetazidine | Sigma-Aldrich | 653322 |
| MitoQ | TRC | TRC-M372215 |
| Isoproterenol | Sigma-Aldrich | I6504 |
| High-fat diet | Envigo | TD.120528 |
| Geltrex | Life technologies | A1413302 |
| Essentisl E8 medium | Gibco | A1517001 |
| Penicillin/streptomycin | Gibco | 15140122 |
| Thiazovivin | Cayman Chemical Company | 004CA14245-25 |
| Chiron | Cayman Chemical Company | 004CA13122-10 |
| RPMI 1640 | Gibco | 61870010 |
| BSA | Sigma-Aldrich | A8412-100ML |
| L-ascorbic acid 2-phosphate | Fujifilm | 321-44823 |
| WNT-C59 | Cayman Chemical Company | 004CA16644-10 |
| B27 | Thermo Fisher Scientific | 17504044 |
| DMEM no gluc/no sod pyr | Gibco | 11966025 |
| Sodium-lactate | Thermo Fisher Scientific | L14500.06 |
| Cryostore | Sigma-Aldrich | C2874-100ML |
| Trypsin | Gibco | 15400054 |
| Dmem high glucose | Thermo Fisher Scientific | 11965092 |
| FBS | Thermo Fisher Scientific | A5256701 |

| Reagent/resource | Reference or source | Identifier or catalog number |
|---|---|---|
| GeneJuice | Millipore | 70967 |
| Puromycin dihydrochloride | Sigma-Aldrich | P8833 |
| EDTA | Sigma-Aldrich | 4005-OP |
| DNeasy Blood & Tissue Kit | QIAGEN | 69504 |
| PCR with LongAmp Taq | NEB | M0323 |
| CNBr–Sepharose beads | GE Healthcare | GE17-0430-01 |
| Proteinase K | Sigma-Aldrich | P2308 |
| Mini and Criterion TGX gels | Biorad | #5678094, #5678093, #4568094, #4568044 |
| chemiluminescent reagent LiteAblot | Euroclone | EMP011005 |
| NADH | Sigma-Aldrich | N8129 |
| $[1-14\,C]$-palmitic acid | PerkinElmer | #NEC075H050UC |
| BCA assay | Sigma-Aldrich | 71285-M |
| L-carnitine | Sigma-Aldrich | C0283 |
| Scintillation liquid UltimaGold | PerkinElmer | 6013321 |
| Cellview cell culture slide, glass bottom | Greiner | 543079 |
| DAPI | Thermo Fisher Scientific | 62248 |
| Prolong Antifade Mounting Medium | Thermo Fisher Scientific | P36984 |
| Duolink® In Situ Red Starter Kit Mouse/Rabbit | Sigma-Aldrich | DUO92101-1KT |
| 6[14 C]-glucose | PerkinElmer | NEC045X050UC |
| BSA-Palmitate Saturated Fatty Acid Complex | Cayman Chemical Company | 29558 |
| Etomoxir | Sigma-Aldrich | 236020 |
| Trizol Reagent | Thermo Fisher Scientific | 15596018 |
| High capacity cDNA reverse transcription kit | Applied Biosystems | 4368814 |
| TaqMan Gene Expression Assays | Applied Biosystems | 4351368 |
| Mouse Malonyl Coenzyme A ELISA kit | CUSABIO | CSB-E12896m |
| Fixable viability dye eFluor 450 | Invitrogen | 65-0863-14 |
| DCFDA-AM | Sigma-Aldrich | 21884 |
| MitoSOX Deep Red | Domino | MT14-12 |
| ATP Bioluminescent Assay Kit | Sigma-Aldrich | FLAA-1KT |
| Bioxytech LPO-586 | OxisResearch | 21012 |
| Sodium Cacodylate | Sigma-Aldrich | C4945 |
| Osmium tetroxide | Electron Microscopy Science | #20816-12-0 |
| Potassium ferrocyanide | Electron Microscopy Science | #13746-66-2 |

| Reagent/resource | Reference or source | Identifier or catalog number |
|---|---|---|
| Uranyl acetate | Electron Microscopy Science | #541-09-3 |
| Embed-812 resin | Electron Microscopy Science | #14900 |
| JC-1 Mitochondrial Membrane Potential Flow Cytometry Assay Kit | Cayman | No. 701560 |
| Luna Universal qPCR sybr Master Mix | NEW ENGLAND BioLabs | M3003E |
| **Software** | | |
| Image Lab software | https://www.bio-rad.com/ | |
| Image J | https://imagej.nih.gov/ij/index.html | |
| GraphPad Prism | https://www.graphpad.com/ | |
| **Other** | | |
| Vevo 2100 Imaging System device | Visualsonics, Inc. | |
| Superose-6 Increase 10/300 GL column | GE Healthcare | |
| AKTA purifier system | GE Healthcare | |
| Chemidoc Touch Imaging System | Biorad | |
| Gas chromatography | Agilent Technologies | |
| High-pressure liquid chromatography HPLC | Jasco | |
| SP8 confocal microscope | Leica | |
| Resipher | Lucid Scientific | |
| ABI Prism 7900HT sequence detection system | Applied Biosystems | |
| QuantStudioTM 6 Real-Time PCR | Thermo Fisher Scientific | |
| FACS Celesta | BD Biosciences | |
| Synergy HTX Multimode reader | Bio-Tec Instruments | |
| TCS SP5 II confocal microscope | Leica | |
| GloMax® Discover Microplate Reader | Promega | |
| Talos 120 C (FEI) electron microscope | Thermo Fisher Scientific | |
| Ceta CMOS camera | Thermo Fisher Scientific | |

## Murine models

Mice received humane care in compliance with Italian law (DL-116, 27 January 1992) and in accordance with the Guide for the Care and Use of Laboratory Animals published by the US National Institutes of Health (NIH Publication No. 85-23, revised 1996). The mice were housed under controlled temperature (20–24 °C) and humidity (50–60%) with a regular 12/12-h light/dark cycle (7 AM–7 PM). They were provided with ad libitum access to food and water and were daily monitored for health status by trained personnel. Male mice 129/Sv wild-type and null for Melusin (Brancaccio et al, 2003) and male mice FVB wild-type and transgenic for human Melusin (De Acetis et al, 2005) of 8–13 weeks of age were used in conformity of the authorizations n. 856/2017-PR 03/11/2017 of the Italian Health Ministry and the guidelines of the University of Turin.

- *Transverse Aortic Constriction*: mice were subjected to transverse aorta constriction (TAC) surgery, as previously described (Lembo et al, 1996). Briefly, mice were anesthetized by intramuscular injection of Zoletil 100 (35 mg/kg) and Rompun (16 mg/kg). A thoracotomy was performed in the second intercostal space to reveal the transverse aorta between the right and left carotid arteries. A 7-0 nylon suture ligature was placed around the aorta and tied against a 27-gauge needle, then promptly removed, to generate a chronic reduction of the aorta lumen. The chest was sutured, and mice were allowed to recover. Controls were subjected to the surgical operation without aortic banding. Four days after surgery, mice were anesthetized with isoflurane to measure the degree of pressure overload by echocardiographic analysis. Only mice with a pressure gradient between 30 mmHg and 60 mmHg were included in the study. Immediately after, the heart was excised and processed for subsequent analysis or frozen in liquid nitrogen.

- *Doxorubicin*:
  In the majority of experiments, a single dose of 4 mg/kg doxorubicin (Sigma-Aldrich) were administered through intra-peritoneal injection. Mice were sacrificed 6 h later, and hearts were excised and processed for subsequent analysis or frozen in liquid nitrogen
  When indicated, a multiple dose of doxorubicin (Sigma-Aldrich) was administered to mice to mimic the human therapeutic regimen (Zhao et al, 2010). Mice received 3 intraperitoneal injections of doxorubicin 4 mg/kg at days 0, 7, and 14 and were followed by echocardiographic analyses for survival. The control group received only a physiologic solution with the same timing.

- *Trimetazidine–protocol 1*: TMZ (Sigma-Aldrich) 30 mg/kg dissolved in physiologic solution was administered orally for 5 days (Ma et al, 2016). The control group received only a physiologic solution.

- *Trimetazidine–protocol 2*: TMZ 0,7 g/L was administered dissolved in drinking water to mice for the duration of the experiment (28 weeks) (Fang et al, 2012). Control group received normal drinking water.

- *MitoQ–protocol 1*: 2 μmol MitoQ (TRC) was administered orally to mice daily for 4 days (Goh et al, 2019). Control group received only a physiologic solution.

- *MitoQ–protocol 2*: MitoQ 20 mg/kg was administered intraperitoneally in mice 5 min before the stress challenge (Powell et al, 2015). The control group received only a physiologic solution.

- *Isoproterenol*: treatment was performed as previously described (Penna et al, 2007). Isolated murine hearts were perfused retrogradely by the Langendorff technique with Krebs–Henseleit bicarbonate buffer containing 1 μM *isoproterenol* (Sigma-Aldrich) for 10 min. Control hearts were perfused with physiologic solution only.

- *High-Fat Diet*: mice were fed ab libitum for 4 weeks with a high-fat diet (HFD - TD.120528 Envigo) or conventional diet. After this period, mice were euthanized and hearts excised and processed for subsequent analysis.

Mice were randomly grouped into treated and control groups. Blinding was not possible because of institutional requirements for labeling the transgenic mouse models and treatments given. When possible, experiments were blinded if conducted by collaborators or other group members.

## Cellular models

Male healthy donor-derived WTC11 human induced pluripotent stem cells (hiPSCs) (Coriell Institute for Medical Research) were previously generated by Bruce Conklin's laboratory at the Gladstones Institutes in San Francisco, USA (Kreitzer et al, 2013). hiPSCs were cultured on Geltrex (Life technologies)-coated tissue culture plates in Essential 8 medium (E8; Gibco) with 0.4% penicillin/streptomycin (Gibco). Cells were grown at sub-confluence, passaged using PBS + EDTA 0.5 mM, and replated in E8 with Thiazovivin 2 µM (Cayman Chemical Company). Melusin-null hiPSCs were generated by CRISPR/Cas9-mediated hemizygous deletion of the *ITGB1BP2* promoter, as better discussed below.

hiPSC-derived cardiomyocytes (hiPSC-CMs) were differentiated from hiPSCs, using an established biphasic WNT modulation protocol (Bertero et al, 2019). Briefly, on day -1, subconfluent hiPSCs were primed for differentiation with 1 µM Chiron (Cayman Chemical Company) in E8. On day 0, cells mesoderm induction was achieved by adding RBA medium [RPMI 1640 (Gibco) + BSA (Sigma-Aldrich) 0.5 mg/mL + L-ascorbic acid 2-phosphate (Fujifilm) 213 ug/mL] supplemented with 4 µM Chiron. On day 2, cardiac mesoderm specification was achieved with RBA supplemented with 2 µM WNT-C59 (Cayman Chemical Company). On day 4, cells were fed with RBA alone. From day 6, cells were cultured with RPMI-B27 medium [RPMI 1640 supplemented with B27 (Thermo Fisher Scientific) and 0.4% penicillin/streptomycin]. Beating cardiomyocytes were observed from day 8. On day 16, hiPSC-CMs were dissociated and plated on Geltrex-coated dishes at a density of $3 \times 10^5$ cells/cm². Cells were lactate-selected for 4 days by feeding with DMEM no glucose/sodium pyruvate (Gibco) with 4 mM sodium L-lactate (Thermo Fisher Scientific). Day 25–27 hiPSC-CMs were harvested for experimental analysis and flow cytometry. Melusin overexpressing hiPSC-CMs were obtained by transduction with AAV6 and AAV9 encoding the *ITGB1BP2* cDNA with a myc tag, while a GFP or empty AAV was used as control.

hiPSCs and hiPSC-CMs were tested every 2 weeks for mycoplasma contamination by PCR.

The human cardiomyocyte cell line (AC16) (#SCC109, Millipore) was cultured in DMEM medium (Gibco) containing 12% FBS (Gibco) and 1% penicillin/streptomycin (Thermo Fisher Scientific). AC16 cells expressing Melusin were obtained by infection with a pLVX lentiviral vector encoding the human *ITGB1BP2* cDNA with a myc-tag. An empty pLVX lentiviral vector was used as a control. Differentiation was favored using the mitogen-free medium, as suggested by the manufacturer. AC16 cells were tested every 2 weeks for mycoplasma contamination by PCR.

- *Doxorubicin*: doxorubicin (Sigma-Aldrich) treatment was performed in line with data published in literature (Yuan et al, 2016).

Cells were grown to 80% confluence and treated with doxorubicin 250 nm or normal media for 24 h.

## Molecular cloning

We generated the CRISPR/Cas9 plasmids needed to knockout Melusin in hiPSCs (pX459-V2.0_ITGB1BP2_-302 and pX459-V2.0_ITGB1BP2_ + 135) according to an established procedure. Briefly, single guide RNAs (sgRNAs) were constructed by cloning 20-mer crispr RNAs (crRNAs) as double-stranded oligonucleotides in pSpCas9 (BB)-2A-Puro (PX459) V2.0 (a gift from Feng Zhang; Addgene plasmid #62988). Two crRNAs were cloned, with PAM sites mapping 302 bp upstream and 135 bp downstream of the transcriptional start site (TSS) of ITGB1BP2: 5′- GCTTAAT CCTCATCGAGCTG-3′ and 5′-GACCCTAATACCAACCTTCC-3′, respectively.

## Generation of Melusin-null hiPSCs

hiPSCs knockout for Melusin were obtained through CRISPR/Cas9-mediated deletion of the promoter and a portion of the first exon of the *ITGB1BP2* gene, similar to what we previously described (Bertero et al, 2019). About $3 \times 10^5$ WTC11 hiPSCs were seeded in a six-well Geltrex-coated dish in E8 supplemented with 2 µM Thiazovivin, and immediately transfected with 2 µg of sgRNA-containing plasmids using 6 µL of GeneJuice (Millipore). The procedure co-delivered 1 µg for each of two sgRNAs targeting the *ITGB1BP2* promoter up- and down-stream: pX459-V2.0_ITGB1BP2_-302 and pX459-V2.0_ITGB1BP2_ + 135, respectively. After 16 h, cells were rinsed in PBS, and fed with E8 containing 2 µM Thiazovivin and 0.5 µg/mL puromycin dihydrochloride (Sigma-Aldrich) to enrich for transfected hiPSCs. After 24 h, the media was replaced with conventional E8 and changed daily. After 72 h, hiPSCs were dispersed to single cells using PBS 0.5 mM EDTA and seeded at a low density of $5 \times 10^3$ cells per 100 mm culture dish in E8 with 2 µM Thiazovivin. After daily media changes with E8 for 7–10 days, pseudo-clonal lines were isolated by picking well-separated individual colonies and expanded for genotyping. Genomic DNA was extracted using the QIAGEN DNeasy Blood & Tissue Kit and used as a template for PCR with LongAmp Taq (NEB). *ITGB1BP2* promoter deletion was assessed using primers located 591 bp and 220 up- and down-stream to the *ITGB1BP2* TSS (fw 5′-GCAGTAATGCAGCAT-CAACCT-3′ and rev 5′-GGGCAGGTAGTTTCTTTCCCC-3′). The resulting product was run on a 1% Agarose-TBE gel and detected by SYBR Safe staining, and two clones (clone 1 and clone 2) with hemizygous deletion of the ~0.4 kb *ITGB1BP2* promoter fragment were isolated. Both Melusin knockout hiPSC lines maintained a normal karyotype throughout the procedure, as assessed by standard G-banding.

## Generation of Melusin over-expressing hiPSCs

hiPSC-CMs were differentiated and lactate-selected. On day 29, cardiomyocytes were washed twice with PBS (w/Ca²⁺ and Mg²⁺) and incubated with the adeno-associated viral vectors (AAV vectors) diluted in RPMI-B27 at a multiplicity of infection (MOI) of 100,000. The AAV vectors used are the following: AAV6-Melusin and AAV9-Melusin and relative controls GFP or empty,

depending on the assay. AAVs for Melusin contain the pAAV-MCS plasmid carrying the cDNA for Melusin, preceded by a myc tag. GFP AVVs contain the pAAV-MCS plasmids carrying the GFP cDNA. A high-density AAVs production was obtained in packaging cells AAV-293T (Agilent) (Vitadello et al, 2020). Forty-eight hours after the infection cells were washed twice with PBS (w/Ca$^{2+}$ and Mg$^{2+}$) and fed with fresh RPMI-B27. On days 34–35, cardiomyocytes were utilized for experimental assays.

## Echocardiography

Echocardiographic analyses were performed on mice treated with doxorubicin, and relative controls, to follow the cardiac parameters and on mice subjected to TAC, and relative controls, to measure the degree of pressure load and follow the cardiac parameters. Mice anesthetized with isoflurane were analyzed using a Vevo 2100 Imaging System device (Visualsonics, Inc.). Two-dimensional-guided M-Mode echocardiography from the parasternal long-axis view was used to measure cardiac parameters. Fractional shortening (FS %) and ejection fraction (EF %) were calculated according to standard formulas. The degree of narrowing of the constricted aorta was evaluated via pulse wave Doppler echography downstream the constriction. Only mice with a pressure gradient between 30 mmHg and 60 mmHg were used for analyses.

## Co-immunoprecipitation assay

Wild-type and Mel null mice were sacrificed to collect their hearts. Hearts were homogenized in IP buffer (25 mM Tris-HCl pH 8, 1 mM MgCl$_2$, 10% glycerol) added with proteases and phosphatases inhibitors and centrifuged at $16,000 \times g$ for 10 min. Supernatants were quantified, and 40 µg of total extract per sample were diluted in Laemmli buffer and stored as input. About 1 mg of total cardiac extract was incubated overnight at 4 °C with 30 µl of anti-Melusin antibody conjugated to CNBr–Sepharose beads (GE Healthcare). The following day, beads were washed in the IP buffer and resuspended in Laemmli buffer.

## Subsarcolemmal and intermyofibrillar mitochondrial purification

Mice wild-type, null or over-expressing Melusin were sacrificed and the hearts were excised and homogenized in isolation buffer (100 mM KCl, 50 mM Tris pH 7.4, 5 mM MgSO$_4$7H$_2$O, 1 mM EDTA, 1.8 mM ATP, 0.5% BSA pH 7.4) added with proteases and phosphatases inhibitors in a glass potter tissue grinder. Samples were clarified by centrifugation at $400 \times g$ for 5 min at 4 °C and the first supernatant and the first pellet were collected separately.

The first supernatant was used to purify subsarcolemmal mitochondria (SS). This was centrifuged at $9000 \times g$ for 10 min at 4 °C obtaining the second supernatant, collected as cytosolic fraction for fractionation experiments. The SS mitochondrial pellet was washed in washing buffer (100 mM KCl, 50 mM Tris pH 7.4, 5 mM MgSO$_4$7H$_2$O, 1 mM EDTA pH 7.4) and resuspended in mitochondrial buffer (0,25 M sucrose, 20 mM Hepes, 1 mM EDTA pH 7.4).

The first pellet was used to purify intermyofibrillar mitochondria (IMF). The pellet was exposed to 5 µg/g trypsin for 10 min in agitation. The pellet was then diluted in lysis buffer and centrifuged at $5000 \times g$ for 5 min at 4 °C. The second supernatant was discarded, while the second pellet was resuspended in lysis buffer and centrifuged at $800 \times g$ for 10 min at 4 °C twice. The supernatants were combined in a new tube and centrifuged at $9000 \times g$ for 10 min at 4 °C to obtain the final IMF mitochondrial pellet resuspended in mitochondrial buffer (0.25 M sucrose, 20 mM Hepes, 1 mM EDTA pH 7.4).

For fractionation experiments, the SS and IMF final mitochondrial pellets were quickly incubated in 2 mg/ml Proteinase K, to digest residues of cytosol, washed in washing buffer and lysed for western blotting analyses.

## Proteinase K accessibility assay

Mitochondria were subjected to treatments with proteinase K, osmotic shock, and Triton X-100 (TX100) to progressively digest mitochondrial compartments. The SS final mitochondrial pellet was washed with washing buffer, resuspended in mitochondrial buffer (0.25 M sucrose, 20 mM Hepes, 1 mM EDTA pH 7.4), and divided in aliquots of 20–30 µg:

Negative controls were resuspended directly in Laemmli buffer for western blot analysis.

Mitochondria treated with osmotic shock (OS) only were resuspended in hypotonic buffer (5 mM sucrose, 2 mM Hepes pH 7.4) 1:40 with respect to the volume of mitochondria for 15 min at 4 °C in agitation. They were centrifuged at $3000 \times g$ for 15 min at 4 °C and resuspended in Laemmli buffer for western blot analysis.

Mitochondria treated with Proteinase K (PK) only were resuspended in 100 µl mitochondrial buffer added with PK 100 µg/ml for 30 min at 4 °C in agitation.

Mitochondria treated with both PK and OS were resuspended 1:40 in a hypotonic buffer for 15 min at 4 °C in agitation. They were centrifuged at $3000 \times g$ for 15 min at 4 °C and then added with PK 100 µg/ml for 30 min at 4 °C in agitation.

Mitochondria treated with PK and TX100 were resuspended in 100 µl mitochondrial buffer added with PK 100 µg/ml and 1% TX100 for 30 min at 4 °C in agitation.

For all the PK treated mitochondria, proteins were precipitated in 10% final volume of TCA for 15 min, centrifuged at $18,000 \times g$ for 20 min at 4 °C and resuspended in Laemmli buffer for western blot analysis.

## Gel filtration chromatography

Gel filtration chromatography was performed on a Superose-6 Increase 10/300 GL column (GE Healthcare) using the AKTA purifier system (GE Healthcare). Wild-type and Mel null mice were sacrificed, and the hearts were excised and homogenized in gel filtration buffer (25 mM Tris-HCl pH 8.0, 1 mM MgCl$_2$, 10% glycerol) added with proteases and phosphatases inhibitors. Extracts were centrifuged at $16,000 \times g$ for 20 min and diluted to 2 mg/ml. About 600 µl of extract was loaded into the machine and separated at a flow rate of 0.5 ml/min. Fractions of 0.5 ml were collected. Molecular mass standards (GE Healthcare) were used to calibrate the column: thyroglobulin (669 kDa), ferritin (440 kDa), catalase (240 kDa), and aldolase (158 kDa). The fractions were

concentrated using Nanosep 3K and resuspended in Laemmli buffer for western blot analyses.

## Western blot

Fresh heart samples and mitochondrial samples were prepared as described above.

Frozen hearts were powdered and homogenized in TBS + 1% TX100 with proteases and phosphatase inhibitors.

Cells were washed from the medium and lysed in a plate with TBS + 1% TX100 with protease and phosphatase inhibitors.

Lysates were finally centrifuged twice for 10 min at $16,000 \times g$ at 4 °C and 40 µg of total protein extract was resuspended in Laemmli buffer. Samples were boiled at 95 °C, separated by electrophoresis in reducing and denaturing polyacrylamide gels, and transferred onto nitrocellulose membranes. The loading was tested by Red Ponceau staining or Coomassie Blue staining. After saturation with TBS, 0.3% Tween 20 added with 5% BSA, blot strips were incubated overnight with primary antibodies and after with the proper secondary antibody. Signals were detected by the chemiluminescent reagent LiteAblot (Euroclone) at the Chemidoc Touch Imaging System (Bio-Rad). Band intensities were quantified using the Image Lab software (Bio-Rad).

Western blotting were performed using the following primary and secondary antibodies: anti-Melusin mouse monoclonal antibody (clone C3, 1:1000 from stock 1 µg/µl) generated in our Institute (Sbroggio et al, 2011), anti-vinculin (Sigma-Aldrich, SAB4200080, 1:1000), anti-HADHA (α-MTP) (Abcam, ab203114, 1:1000), anti-HADHB (β-MTP) (Santa Cruz Biotechnology, sc-365907, 1:1000), anti-VDAC1 (Abcam, ab15895, 1:1000), anti-actin (Santa Cruz Biotechnology, sc-1616-R, 1:1000), anti-tom20 (Santa Cruz Biotechnology, sc-11415, 1:1000), anti-total OXPHOS WB antibody cocktail (Abcam, ab110413, 1:1000), anti-pyruvate kinase PKM2 (Thermo Fisher Scientific, MA5-14890, 1:1000), anti-phospho-pyruvate dehydrogenase α1 (ser293) (Cell signaling, 31866S, 1:1000), anti-pyruvate dehydrogenase (Cell signaling, 3205, 1:1000), anti-catalase (Abcam, ab16731, 1:1000), anti-glutathione peroxidases 1 (Thermo Fisher Scientific, #PA5-26323, 1:1000), anti-glutathione peroxidases 4 (Santa Cruz Biotechnology, sc-166570, 1:1000), anti-peroxiredoxin 1 (Abcam, ab109498, 1:1000), anti-peroxiredoxin 3 (Abcam, ab222807, 1:1000), anti-superoxide dismutase 1 (Santa Cruz Biotechnology, sc-11407, 1:1000), anti-superoxide dismutase 2 (Santa Cruz Biotechnology, sc-133134, 1:1000), membrane integrity wb antibody cocktail (Abcam, ab110414, 1:1000), anti-Nox4 (Thermo Fisher Scientific, #14347-1-AP, 1:2000), anti-rabbit IgG-peroxidase antibody produced in goat (Sigma-Aldrich, #A6154, 1:5000) and anti-mouse IgG-peroxidase antibody produced in goat (Sigma-Aldrich, #A4416, 1:10,000).

## α-MTP activity

The α-MTP activity was calculated as previously reported (Ljubkovic et al, 2019). About 20 mg of frozen heart powder was lysed in lysis buffer (5 mM Hepes, 1 mM EGTA, 1 mM DTT, 0.1% TX100, pH 8.7) and clarified by centrifugation 1 min at $1 \times g$ at 4 °C. Samples were quantified spectrophotometrically and 40 to 400 µg were aliquoted in Eppendorf. Samples were added with 1 ml of reaction buffer (100 mM Na-pyrophosphate, 1 mM EDTA, 0.15 mM NADH, 50 µM acetoacetyl-CoA) and 300 µl were aliquoted in a 96-well plate. The plate was read at $\lambda = 340$ nm for 160 cycles, and the $V_{max}$ was calculated by linear regression on the exponential portion of the curve.

## Palmitate oxidation capacity

Palmitate oxidation capacity was evaluated in mitochondria isolated from cardiac tissues and cells using $[1-{}^{14}C]$-palmitic acid (PerkinElmer). Hearts were homogenized in isolation buffer (Tris-HCl 50 mM, KCl 100 mM, MgCl$_2$ 5 mM, EDTA 1 mM, ATP 1.8 mM, pH 7.2) added with proteases and phosphatases inhibitors. For cells, the medium was removed, and cells were washed with PBS and gently detached by scraping in buffer added with proteases and phosphatase inhibitors. Samples were homogenized in a glass potter tissue grinder and centrifuged at $400 \times g$ for 1 min at 4 °C to remove nuclei and other large membranous organelles. Supernatants were further centrifuged at $16,000 \times g$ for 5 min at 4 °C. The pellet enriched in mitochondria was resuspended in 250 µl resuspension buffer (Sucrose 250 mM, K$_2$HPO$_4$ 15 mM, MgCl$_2$ 2 mM, EDTA 0.5 mM, pH 7.2) for the analysis. A small aliquot was used for protein quantification by BCA assay (Millipore). About 50 µl of samples was incubated with 100 µl of assay buffer (20 mM Hepes, 0.24 mM fatty acid-free BSA, 0.5 mM L-carnitine) and 2 µCi (380 µM) $[1-{}^{14}C]$-palmitic acid (PerkinElmer) for 1 h in mild agitation at RT. About 100 µl of a solution 1:1 v/v phenylethylenamine 100 mM/methanol was added and incubated for 1 h further at room temperature. The reaction was stopped with 100 µl of HClO$_4$ 0.8 N and centrifuged at $16,000 \times g$ for 10 min. Supernatants (containing $^{14}CO_2$, used as internal negative control) and pellets (containing $^{14}C$-acid soluble metabolites (ASM)) were mixed with 4 ml of scintillation liquid (UltimaGold, PerkinElmer) and read by a β-counter. In each experimental condition, the amount of $^{14}C$-$CO_2$ was always ≤5% of $^{14}C$-ASM. Results were expressed as picomoles of $^{14}C$-ASM per hour normalized on mg of proteins.

## Lipidomic analyses

Lipid composition analysis was performed extracting lipids in chloroform/methanol mixture and partitioned with water. The organic phase was dried and then suspended in chloroform/methanol for the analysis of total fatty acid and cholesterol contents. Total fatty acids were determined as methyl esters by gas chromatography (Agilent Technologies). The cholesterol analysis was achieved by high-pressure liquid chromatography HPLC (Jasco).

## Immunofluorescence

AC16 cells empty and mel were plated in cellview cell culture slide, glass bottom, (Greiner) at concentration of 15,000 cells/well. For hiPSC-CMs, cellview cell culture slide was coated with poly-L lysine and ES-qualified Geltrex (0.17 mg/ml). Mel null hiPSC-CMs were plated at a concentration of 30,000 cells/well and infected with 30,000 MOI AAV6-empty and AAV6-MycMel as described above. hiPSC-CMs were used 7 days after the infection. Cells were fixed with 4% PFA for 10 min at RT, permeabilized with 0.1% TX100 in PBS, and washed in TBS + 10% Tween 20 (TBST). Cells were blocked with PBS + 3% BSA + 1% goat serum for 1 h and incubated 1 h at RT in a humidity chamber with a solution of rabbit anti-HADHA (α-MTP) (Abcam, ab203114) diluted 1 µg/ml or mouse anti-Myc tag (Thermo Fisher Scientific, MA1-21316) diluted 5 µg/ml in PBS + 3% BSA + 1% goat serum. Anti-myc tag was used since no antibodies for wild-type Melusin work in IF in

our hands. Cells were washed three times for 5 min with PBS and incubated at RT in a humidity chamber for 1 h with a dilution of goat anti-rabbit secondary antibody Alexa Fluor™ 633 (Thermo Fisher Scientific, A21070) 4 µg/ml in PBS + 1% BSA or goat anti-mouse secondary antibody Alexa Fluor™ 488 (Thermo Fisher Scientific, A11001) 1 µg/ml in PBS + 1% BSA. Negative controls were performed by staining cells with no primary antibody and secondary antibody only. Cells were washed three times for 5 min with PBS and incubated in a humidity chamber for 1 min at RT with a dilution of 0.1 µg/ml of DAPI (Thermo Fisher Scientific) in deionized water for nuclear staining. Cells were washed three times for 5 min with PBS, and cover slips were mounted using the Prolong Antifade Mounting Medium (Thermo Fisher Scientific). Images were acquired using the 63x objective of Leica SP8 confocal microscope at ex:488 nm/em:520 nm for the anti-mouse secondary antibody Alexa Fluor™ 488, at ex:631 nm/em:650 nm for the anti-rabbit secondary antibody Alexa Fluor™ 633.

## Proximity ligation assay

Proximity ligation assay between Melusin and α-MTP was performed using the Duolink® In Situ Red Starter Kit Mouse/Rabbit (Sigma-Aldrich), following the manufacturer's recommendations. AC16 cells empty and mel were plated in cellview cell culture slide, glass bottom, (Greiner) at concentration of 15,000 cells/well. For hiPSC-CMs, cellview cell culture slide was coated with poly-L lysine and ES-qualified Geltrex (0.17 mg/ml). Mel null hiPSC-CMs were plated at a concentration of 30,000 cells/well and infected with 30,000 MOI AAV6-empty and AAV6-MycMel as described above. hiPSC-CMs were used 7 days after the infection. Cells were fixed with 4% PFA for 10 min at RT, permeabilized with 0.1% TX100 in PBS, and washed in TBS + 10% Tween 20 (TBST). Cells were blocked with Duolink® Blocking Solution and incubated for 1 h at 37 °C in a preheated humidity chamber with a solution of rabbit anti-HADHA (α-MTP) (Abcam, ab203114) and mouse anti-Myc tag (Thermo Fisher Scientific, MA1-21316) diluted respectively 1 µg/ml and 5 µg/ml in Duolink® Antibody Diluent. Cells were washed twice for 5 min with Wash Buffer A at RT and incubated in a preheated humidity chamber for 1 h at 37 °C with a dilution of PLA PLUS and MINUS probes 1:5 in Duolink® Antibody Diluent. After washes, cells were incubated in a preheated humidity chamber for 30 min at 37 °C with a solution 1:40 of ligase in Duolink® Ligation Buffer. Cells were washed and incubated in a preheated humidity chamber for 100 min at 37 °C with a solution 1:80 of polymerase in Amplification Buffer. Cells were washed twice for 10 min with Wash Buffer B at RT and the chambers and silicone around the wells were removed from the slide completely. Cover slips were mounted with the Duolink® In Situ Mounting Medium with DAPI and the edges of the cover slip were sealed with nail polish. PLA images were acquired at least 20 min after using the 63x objective of Leica SP8 confocal microscope at ex:594 nm/em:624 nm.

## Glucose flux through a tricarboxylic acid cycle

The glucose flux through the tricarboxylic acid cycle was analyzed on living cells using 6[$^{14}$C]-glucose (PerkinElmer). The cell medium was removed, and cells were washed in PBS and detached by gently scraping in PBS. Cells were centrifuged at $100 \times g$ for 5 min and resuspended at $2 \times 10^5$ cells in 1 ml of Hepes-Ca$^{++}$ buffer (145 mM NaCl, 5 mM KCl, 1 mM MgSO$_4$, 10 mM Hepes sodium salt, 10 mM glucose, 1 mM CaCl$_2$, pH 7.4). A small aliquot was used for protein quantification by BCA assay. Cells were transferred in a vial and added with 2 µCi (80 µM) of 6[$^{14}$C]-glucose. The vial, connected with a tight connector closed by a rubber cap with another vial containing 0.5 ml KOH 0.8 N, was incubated for 1 h at 37 °C in agitation. Later, 0.4 ml of HClO$_4$ 3.7 N was added to the cell suspension by a syringe to stop the reaction. The system was adjusted to have sample vials at 37 °C and the KOH vials in ice, in order to maximize the reaction of $^{14}$CO$_2$ produced by the TCA cycle with the KOH, and maintained 1 h in agitation. The so generated [$^{14}$C]-K$_2$CO$_3$, index of the TCA cycle activity, was mixed with 4 ml scintillation liquid (UltimaGold, PerkinElmer) and read by a β-counter. The marked $^{14}$CO$_2$ produced via the tricarboxylic acid cycle was expressed as picomoles CO$_2$ produced per hour normalized on the mg of cell proteins.

## Oxygen consumption rate

The oxygen consumption rate (OCR) of cells expressing or not Melusin was evaluated using the Resipher (Lucid Scientific) technology.

hiPSC-CMs wild-type and null for Melusin were plated overnight at near confluence (90,000 cells per well) in a 96-well plate in limited medium (DMEM glucose - glutamine - pyruvate -, NaHCO$_3$ 3.7 g/L, 0.5 mM glucose, 2 mM glutamax, 1% FBS, 1% penicillin/streptomycin) added with 1 mM L-carnitine. As negative controls, some wells were added with 40 µM of the carnitine palmitoyltransferase-1 (CPT-1) inhibitor Etomoxir.

AC16 cells empty and mel were plated overnight at the confluence (30,000 cells per well) in a 96-well plate in a limited medium added with 1 mM L-carnitine. As negative controls, wells were added with 40 µM Etomoxir.

The morning after, the plate was attached to the Resipher in the incubator and left to equilibrate for 5 h. The OCR at this time point was considered as time = 0. The plate was then detached from the machine, and a palmitate-BSA conjugate was added to the wells (150 µM for AC16 cells, 1 mM for hiPSC-CMs). The plate was attached again to the Resipher, and the OCR was followed for 25–30 h. The maximum pick of OCR was considered as time = 1 to measure the maximal OCR increase (t = 1 – t = 0).

## RNA isolation and real-time PCR

Total RNA from frozen heart powder or cell cultures was isolated using Trizol Reagent (Thermo Fisher Scientific), following the manufacturer's recommendations. RNA was reverse transcribed by using Applied Biosystem high-capacity cDNA reverse transcription kit. Gene expression analysis was performed using TaqMan Gene Expression Assays (Applied Biosystems) on an ABI Prism 7900HT sequence detection system (Applied Biosystems). 18S (Thermo Fisher Scientific) was used as endogenous control and gene-specific primers were used for other genes (Roche Applied Sciences). Analysis was performed using the $^{\Delta\Delta}$Ct method to determine fold changes. The specific primers used were: *ITGB1BP2* forward 5′- CCTTCCTGATTCCTGTTGCCA-3′, reverse 5′-TCTACAGTTCGCTTTCGGCA-3′; *α-MTP* forward 5′-TTCTTAAAGACACCACAGTGACG-3′, reverse 5′-CTTCTTCACTTTGTCGTTCAGC-3′; *β-MTP* forward 5′-CAAGCAATGTGGCTAGAGAGG-3′,

reverse 5′-AGAGATACAAGCCATGGTGACA-3′; *NONO* forward 5′-AGATGGCTATGGGAGGTGCTA-3′, reverse 5′-AGCCTGACCAA AGCGTTCAG-3′; *HPRT* forward 5′-TGACACTGGCAAAACAAT GCA-3′, reverse 5′-GGTCCTTTTCACCAGCAAGCT-3′; *TNNT2* forward 5′-TTCACCAAAGATCTGCTCCT CGCT-3′, reverse 5′-TTATT ACTGGTGTGGAGTG GGTGTGG-3′; *GLUT1* forward 5′-CCTGT CTCTTCCTACCCAACC-3′, reverse 5′-GCAGGAGTGTCCGTGTC TTC-3′; *GLUT4* forward 5′-CTCATGGGCCTAGCCAATG-3′, reverse 5′-GGGCGATTTCTCCCACATAC-3′;

## Malonyl-CoA ELISA assay

Malonyl-CoA level was measured using the Mouse Malonyl Coenzyme A ELISA kit (CUSABIO), following the manufacturer's recommendations. Briefly, 100 mg of cardiac tissue from wild-type, Mel null and Mel over mice were homogenized in 1 ml of PBS and stored overnight at −20 °C. The day after, samples were centrifuged at 5000×g for 5 min at 4 °C, and the supernatant was collected for the assay. Supernatants were diluted 1:3000 in dilution buffer and 100 µl was added in the wells of a coated assay plate and incubated 2 h at 37 °C. A standard solution was used as calibrator at 10, 5, and 1 ng/ml, while dilution buffer was used as 0. Samples were removed and 100 µl of biotin-antibody 1x was added and incubated 1 h at 37 °C. Wells were washed with wash buffer and incubated with 100 µl of HRP-avidin 1 × 1 h at 37 °C. Wells were washed again and filled with 90 µl of TMB substrate for 15–30 min at 37 °C. About 50 µl of Stop solution was added to the wells before reading the optical density at 450 nm.

## Ratio NADH/NAD$^+$ and NADPH/NADP$^+$

Frozen hearts were lysed at 4 °C in 0.6 M perchloric acid (PCA) for NAD$^+$ and NADP$^+$ determination or in 0.1 M NaOH for NADH and NADPH determination. The lysate was clarified by centrifugation. NAD$^+$/NADP$^+$ contents were quantified as previously described (Bruzzone et al, 2009). NADH/NADPH content was measured by fluorescence, as previously reported (Pittelli et al, 2010).

## Flow cytometry

Live hiPSC-CMs were resuspended in 200 µL of FACS buffer (PBS, 5% FBS, and 0.05% sodium azide) into a 96-well plate and centrifuged at 220×g for 4 min to remove the supernatant. Samples were incubated with 100 µL of fixable viability dye eFluor 450 (Invitrogen) diluted 1:1000 in PBS for 10 min at RT covered from light. Samples were washed in FACS buffer, fixed in PBS + 4% paraformaldehyde (Alfa Aesar) for 15 min at RT, and washed twice in FACS buffer. Cells were then permeabilized and blocked in PBS 0.75% saponin 5% FBS, and stained in 50 µL of PBS 0.75% saponin with anti-cTnT AF647 antibody (BD Pharmingen) diluted 1:100, incubating for 40 min at RT covered from light. An isotype control staining was performed in parallel. The stained samples were centrifuged and washed twice in PBS 0.75% saponin, and finally resuspended in FACS buffer for the analysis using a FACS Celesta. Data analysis was performed using FlowJo. Antibodies used were: Alexa Fluor 647 Mouse Anti-Cardiac Troponin T; clone 13-11 (BD Pharmingen); Alexa Fluor 647 Mouse IgG1, κ Isotype Control; clone MOPC-31 C (BD Pharmingen)

## ROS content

ROS amount was evaluated with two different approaches:

The whole pool of ROS was measured by 5(6)-Carboxy-2′,7′-dichlorofluorescein diacetate (Sigma-Aldrich, DCFDA-AM) in mitochondria isolated from cardiac tissues and cells, processed as described for Palmitate oxidation capacity. A small aliquot was used for protein quantification by BCA assay.

10 µl of the sample was incubated with 10 µM DCFDA-AM for 30 min at 37 °C. The fluorescence was read at ex:504 nm/em:529 nm with a Synergy HTX Multimode reader (Bio-Tec Instruments), using H$_2$O$_2$ concentrations (1 pM–10 µM) as standards. Results were expressed as moles total ROS per mg of proteins.

Mitochondrial superoxide by MitoSOX Deep Red (Dojindo). Cells were plated in culture media in cellview cell culture slide, glass bottom, (Greiner) at a concentration of 8000 cells/well for AC16 empty and Mel and 30,000 cells/well for hiPSC-CMs wild-type and null for Melusin. As positive control, one well was incubated with 1 mM Paraquat overnight, as negative control one well was incubated with 1 µM MitoQ overnight. The day after, cells were incubated with 5 µM MitoSOX and 5 µg/ml of Hoechst staining for nuclei for 30 min in an incubator protected from light. Cells were washed and suspended in culture media. Fluorescence was read using the 20X immersion objective of Leica TCS SP5 II confocal microscope at ex:405 nm/em:415–498 nm for Hoechst and ex:561 nm/em:640–700 nm for MitoSOX. Image J was used to process the images. Results were expressed as RawIntDen/number of nuclei.

## Electron flux assay

Mitochondria isolated as described above and resuspended in mitochondrial buffer were used to measure the electron flux from mitochondrial Complex I to Complex III. In a 96-well plate, 10 µl of mitochondria was incubated with 160 µl buffer A (5 mM KH$_2$PO$_4$, 5 mM MgCl$_2$, 5% BSA) and 100 µl buffer B (50 mM KH$_2$PO$_4$, 5 mM MgCl$_2$, 5% BSA, 25% saponin). Freshly prepared cytochrome c oxidized form 0.12 mM and NaN$_3$ 0.2 mM were added. The plate was left to equilibrate for 1 min at RT, then the reaction was started by adding NADH 0.15 mM. The kinetics of the reduction of cytochrome c was monitored at 550 nm for 6 min, using a Synergy HTX Multimode reader (Bio-Tek Instruments). The velocity rate was extrapolated from the curve, and results were calculated according to the Lambert–Beer equation and expressed as picomoles of cytochrome c reduced/minutes per mg of mitochondrial proteins.

## ATP measurement

The ATP content was measured in both isolated mitochondria and in cardiac total extracts using the ATP Bioluminescent Assay Kit (Sigma-Aldrich) following the manufacturer's recommendations. Mitochondria were isolated as described above and resuspended in mitochondrial buffer. Hearts (15–20 mg) were homogenized in 350 µl of PBS added with proteases inhibitors, boiled 5 min at 95 °C to denature ATPases and centrifuged at 16000×g for 5 min. Supernatants were used for the assay.

In a 96-well plate, 100 µl of ATP assay mix was added with 100 µl of sample or ATP standard as calibrator. Luminescence was immediately measured with a luminometer. Results were expressed as nanomoles of ATP per mg of mitochondrial or whole tissue proteins.

## Lipid peroxidation assay

Lipid peroxidation in cardiac samples was evaluated by detecting malondialdehyde (MDA) and 4-hydroxyalkenals (HAE) using the Bioxytech LPO-586 (OxisResearch, USA), following the manufacturer's recommendations. About 30–40 mg of cardiac tissue was homogenized in 350 µl of PBS added with 5 mM butylated hydroxytoluene (BHT) in acetonitrile and centrifuged at $3000 \times g$ for 10 min at 4 °C. Supernatant was saved for the assay, and a small aliquot was used for protein quantification. About 200 µl of the sample was mixed with 650 µl of N-methyl-2-phenylindole in acetonitrile (Reagent R1) diluted in methanol. About 150 µl of 15.4 M methanesulfonic acid (Reagent R2) was added and incubated for 1 h at 45 °C. Samples were finally centrifuged at $15,000 \times g$ for 10 min, and supernatants were collected to measure the absorbance at 586 nm with a spectrophotometer. A blank with water and standards at different concentrations of MDA were used. Results were calculated as suggested and expressed as nanomoles of MDA + HAE per g of wet weight.

## Transmission electron microscopy (TEM)

Small pieces of cardiac tissue were fixed with 2.5% glutaraldehyde (Sigma-Aldrich) + 100 mM cacodylate (Sigma-Aldrich) and postfixed with 1% osmium tetroxide (Electron Microscopy Science), 1.5% potassium ferrocyanide (Electron Microscopy Science) + 100 mM cacodylate. Samples were stained in 0.5% uranyl acetate (Electron Microscopy Science), dehydrated, and finally embedded in Embed-812 resin (Electron Microscopy Sciences). Blocks were cut in ultrathin sections (70 nm) and contrasted with uranyl acetate (Electron Microscopy Science) and Sato's lead solutions (lead citrate tribasic, Sigma-Aldrich; lead nitrate, Merck; lead acetate trihydrate, Sigma-Aldrich; sodium citrate tribasic dehydrate, Sigma-Aldrich) and observed with Talos 120 C (FEI) electron microscope. Images were acquired with a 4k × 4 K Ceta CMOS camera (Thermo Fisher Scientific). Image J was used to process the images and count the cristae distance.

## Mitochondrial membrane potential measurement

Mitochondrial membrane potential was evaluated using the JC-1 Mitochondrial Membrane Potential Flow Cytometry Assay Kit (Cayman). AC16 cells empty and mel were plated in a 12-well plate at the concentration of 120,000 cells per well. hiPSC-CMs wild-type and null for Melusin were plated in a 12-well plate at the concentration of 500,000 cells per well. Cells were treated with doxorubicin as described above. Cells were detached by trypsinization, resuspended in cell medium, and finally centrifuged at $200 \times g$ for 5 min at RT. Cells were resuspended in 100 µl of assay buffer, excepting one well resuspended in assay buffer added with 50 µM FCCP, as a positive control. After 5 min of incubation at RT, cells were stained with 100 µl of JC-1 2X working solution diluted in culture media. A JC-1 unstained sample was used as a normalizer for doxorubicin auto-fluorescence. Data were immediately acquired at the Celesta FACS machine at ex:405 nm/em:605-308 (Perelman

et al, 2012), analyzed using FlowJo and expressed as a percentage of mitochondria with a physiologic mitochondrial membrane potential compared to unstained controls.

## DNA extraction and mitochondrial DNA quantification

The genomic DNA (gDNA) and mitochondrial DNA (mtDNA) were extracted by DNeasy Blood and Tissue Kit (Qiagen) following the manufacturer's recommendations and quantified using Nanodrop. The genomic and mitochondrial gene expression profiles were tested by Luna Universal qPCR sybr Master Mix (NEW ENGLAND BioLabs) using 10 ng of DNA for mtDNA and 40 ng of DNA for gDNA. The thermal cycling conditions at the QuantStudio™ 6 Real-Time PCR were as follows: 1 cycle of 95 °C for 5 min, followed by 40 cycles of 95 °C for 15 s and 60 °C for 1 min (Kumpunya et al, 2022). The relative content of mtDNA was normalized to genomic β-2-microglobulin ($\Delta ct$ = gDNA ct – mtDNA ct) and determined by the $2 \times 2^{\Delta ct}$. Wildtype, Mel null and Mel over, untreated and treated, data were normalized on the media of the respective untreated wild-type control. The primer sequences used were: mtDNA: 5′-CGTACACCCTCTAACCTAGAGAAGG-3′, 5′-GGTTTTAAGTCTTACGCAATTTCC-3′ and β2m: 5′-TTCTGGTGC TTGTCTCACTGA -3′, 5′- CAGTATGTTCGGCTTCCCATTC-3′.

## Statistical analyses

Statistical analyzes were performed using GraphPad Prism 9. The number of biological replicates is indicated in the Fig. legend, and data were presented as individual data points overlayed with means ± SD. The normal distribution of data was tested by Shapiro–Wilk tests, to determine whether parametric statistical tests could be employed. Depending on the variables involved, significance was tested using the unpaired two-tailed Student's t-test, one-way ANOVA, or two-way

### The paper explained

**Problem**

Cardiac resilience is the ability of the heart to cope with stress. Defining the molecular mechanisms promoting resilience can allow us to define effective therapeutic approaches to treat cardiomyopathies. The heart relies mainly on mitochondrial FAO to sustain its energetic request, at the cost of higher production of ROS. When stressful events further raise ROS levels, mitochondrial dysfunction may occur, causing maladaptive remodeling and heart failure.

**Results**

We discovered that the cardioprotective chaperone Melusin binds to and inhibits the MTP, a key enzyme of FAO in mitochondria. We revealed that the tuning of FAO mediated by Melusin restrains ROS levels in physiologic conditions and further hampers the ROS boost during stress, protecting the heart from mitochondrial dysfunction and functional impairment. The administration of the MTP inhibitor, trimetazidine, before or concomitantly to the stressful challenge achieves similar results, promoting cardiac resilience to pathologic stimuli.

**Impact**

Our results demonstrate that limiting FAO in anticipation of cardiotoxic treatments or timely during myocardial stressful events may represent an effective approach to protect the heart from maladaptive remodeling and to ameliorate the condition of cardiopathic patients.

ANOVA with Tukey/Bonferroni post hoc correction, as detailed in the relative Fig. legend. For experiments in which we evaluated the amplitude of difference among one genotype and its control set to 100%, we performed a "one sample $t$-test" comparing the mean of the group to the hypothetical mean represented by 100% of the control group. A $p$ value below 0.05 was considered to be statistically significant. The Mantel–Cox test was used in Kaplan–Meier survival curves.

## Data availability

This study includes no data deposited in external repositories.

The source data of this paper are collected in the following database record: biostudies:S-SCDT-10_1038-S44321-024-00132-z.

## Peer review information

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

## Acknowledgements

We thank Prof. Paolo Porporato and Alessio Menga for their precious help in metabolic analysis. We thank Federica Logrand and James Cimino, for their technical assistance in echography analysis, and Tiziana Cravero and Cristina Rubinetto for technical assistance in antibody preparation. We are grateful to Bruce Conklin for sharing the WTC11 hiPSC line.

This work was supported by Italian Ministry of Education, University and Research (Progetto di Ricerca PRIN 2015 P2015 2015EASE8Z and Progetto di Ricerca PRIN 2022 P20225EMBT), Compagnia di San Paolo/University of Torino (Progetti di ricerca di Ateneo 2016, project number CSTO165798) to MB, by a grant from the National Heart, Lung and Blood Institute of the National Institutes of Health (R01 HL149734) to AB and NJS and by Associazione Italiana per la Ricerca sul Cancro (IG21408, IG29250) to CR.

## Author contributions

**Matteo Sorge**: Conceptualization; Formal analysis; Supervision; Validation; Investigation; Visualization; Methodology; Writing—original draft; Project administration; Writing—review and editing. **Giulia Savorè**: Conceptualization; Resources; Formal analysis; Validation; Investigation; Methodology; Writing—original draft. **Andrea Gallo**: Investigation; Methodology. **Davide Acquarone**: Investigation; Writing—original draft; Writing—review and editing. **Mauro Sbroggiò**: Formal analysis; Investigation; Methodology. **Silvia Velasco**: Formal analysis; Investigation; Methodology. **Federica Zamporlini**: Formal analysis; Investigation. **Saveria Femminò**: Investigation. **Enrico Moiso**: Investigation. **Giampaolo Morciano**: Investigation. **Elisa Balmas**: Resources; Methodology. **Andrea Raimondi**: Investigation. **Gabrielle Nattenberg**: Methodology. **Rachele Stefania**: Resources. **Carlo Tacchetti**: Resources; Investigation. **Angela, Maria Rizzo**: Investigation. **Paola Corsetto**: Investigation. **Alessandra Ghigo**: Resources; Methodology; expert consulent. **Emilia Turco**: Resources; Methodology; expert consulent. **Fiorella Altruda**: expert consulent. **lorenzo silengo**: expert consulent. **Paolo Pinton**: Investigation; expert consulent. **Nadia Raffaelli**: Resources; Investigation. **Nathan, J Sniadecki**: Resources; Methodology. **Claudia Penna**: Investigation; Methodology. **Pasquale Pagliaro**: Investigation; Methodology. **Emilio Hirsch**: Writing—original draft; Writing—review and editing; expert consulent. **Chiara Riganti**: Resources; Funding acquisition; Investigation; Methodology; Writing—original draft; Writing—review and editing. **Guido Tarone**: Conceptualization; Supervision; Funding acquisition; Project administration. **Alessandro Bertero**: Conceptualization; Resources; Funding acquisition; Methodology; Writing—original draft; Writing—review and editing. **Mara Brancaccio**: Conceptualization; Resources; Supervision; Funding acquisition; Validation; Writing—original draft; Project administration; Writing—review and editing.

Source data underlying figure panels in this paper may have individual authorship assigned. Where available, figure panel/source data authorship is listed in the following database record: biostudies:S-SCDT-10_1038-S44321-024-00132-z.

## Disclosure and competing interests statement

The authors declare no competing interests.

# Expanded View Figures

**Figure EV1.  Melusin binds the mitochondrial trifunctional protein.**

(A) Coomassie Blue staining of Melusin immunoprecipitation (IP) from cardiac total extracts of Mel over mice and Mel null mice as controls. The indicated bands were cut out and identified by mass spectrometry. (B) Melusin immunoprecipitation from cardiac total extracts of wild-type mice and Mel null mice, as controls. The immunoprecipitated Melusin and the co-immunoprecipitated α-MTP and β-MTP were detected by immunostaining. Vinculin was stained to verify the complete removal of the total extract. Representative of $n = 3$ independent experiments. (C) Mitochondrial (Mito) and cytosolic (Cyto) fractions are differentially isolated from wild-type hearts. Melusin presence was evaluated by immunostaining. Vinculin and actin were stained as cytosolic markers, α-MTP and Vdac1 as mitochondrial markers. Representative of $n = 4$ independent experiments. (D) Mitochondria isolated from wild-type hearts treated with Proteinase K (PK), osmotic shock (OS), and Triton X-100 (TX100), or a combination of them, in order to digest proteins of different mitochondrial compartments. Melusin presence was evaluated by immunostaining. Pdh and α-MTP were stained as markers of the matrix compartment, Vdac1 as markers of the outer mitochondrial membrane. Representative of $n = 3$ independent experiments. (E) Fractions of cytosol (Cyto), subsarcolemmal (SS), and intermyofibrillar (IMF) mitochondria differentially isolated from wild-type hearts. Melusin presence was evaluated by immunostaining. α-MTP and Vdac1 were stained as mitochondrial markers. Representative of $n = 3$ independent experiments. (F) Complexes of different molecular weights were obtained as gel filtration fractions from the cardiac total extract of wild-type, Mel null and Mel over mice. Fractions were immunostained for α-MTP, β-MTP, and Melusin. Red arrows indicate the corresponding molecular weights. Representative of $n = 3$ independent experiments. IP immunoprecipitation, PK proteinase K, OS osmotic shock, Vinc vinculin, Vdac1 voltage-dependent anion-selective channel 1, Pdh pyruvate dehydrogenase, SS subsarcolemmal mitochondria, IMF intermyofibrillar mitochondria.

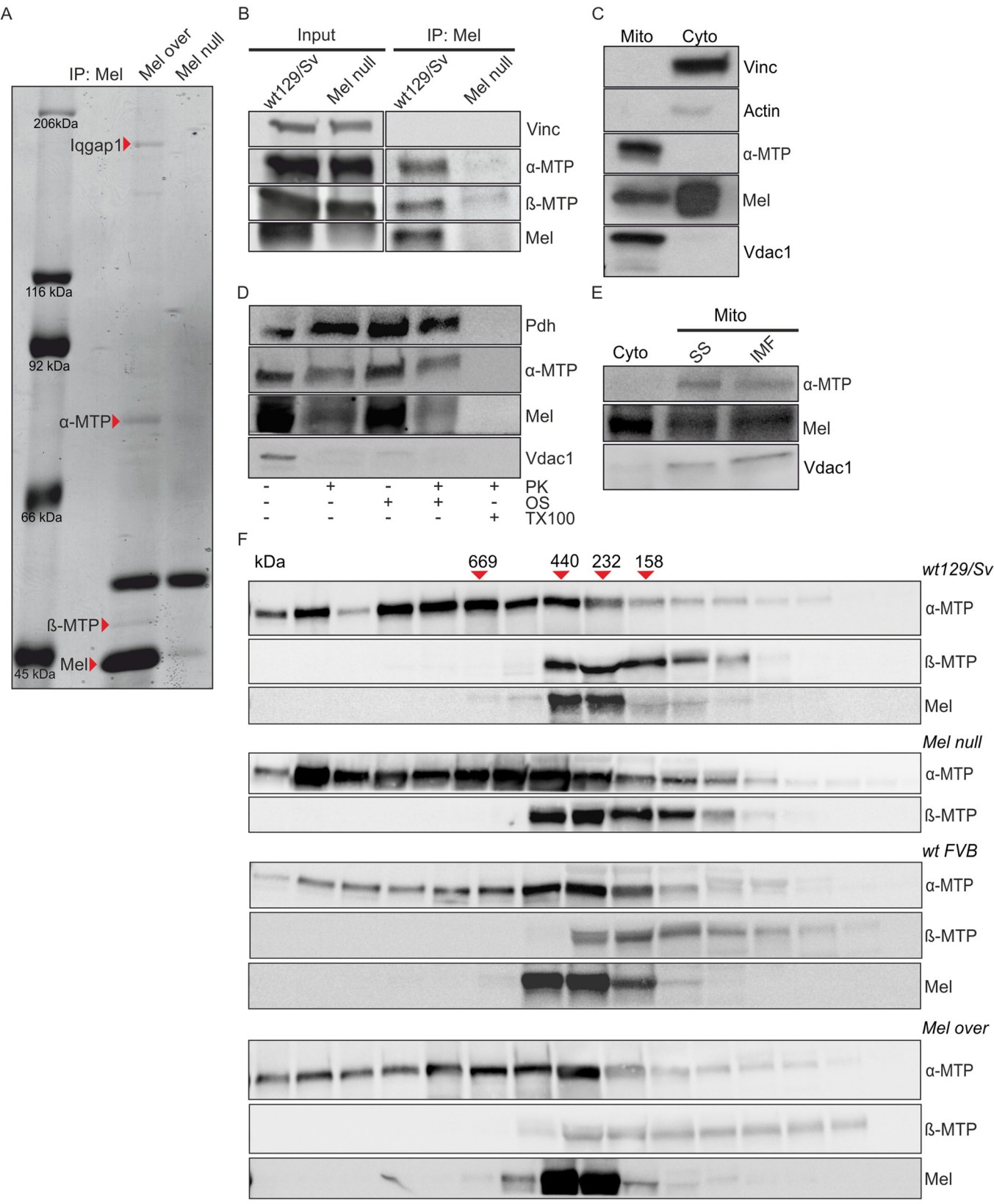

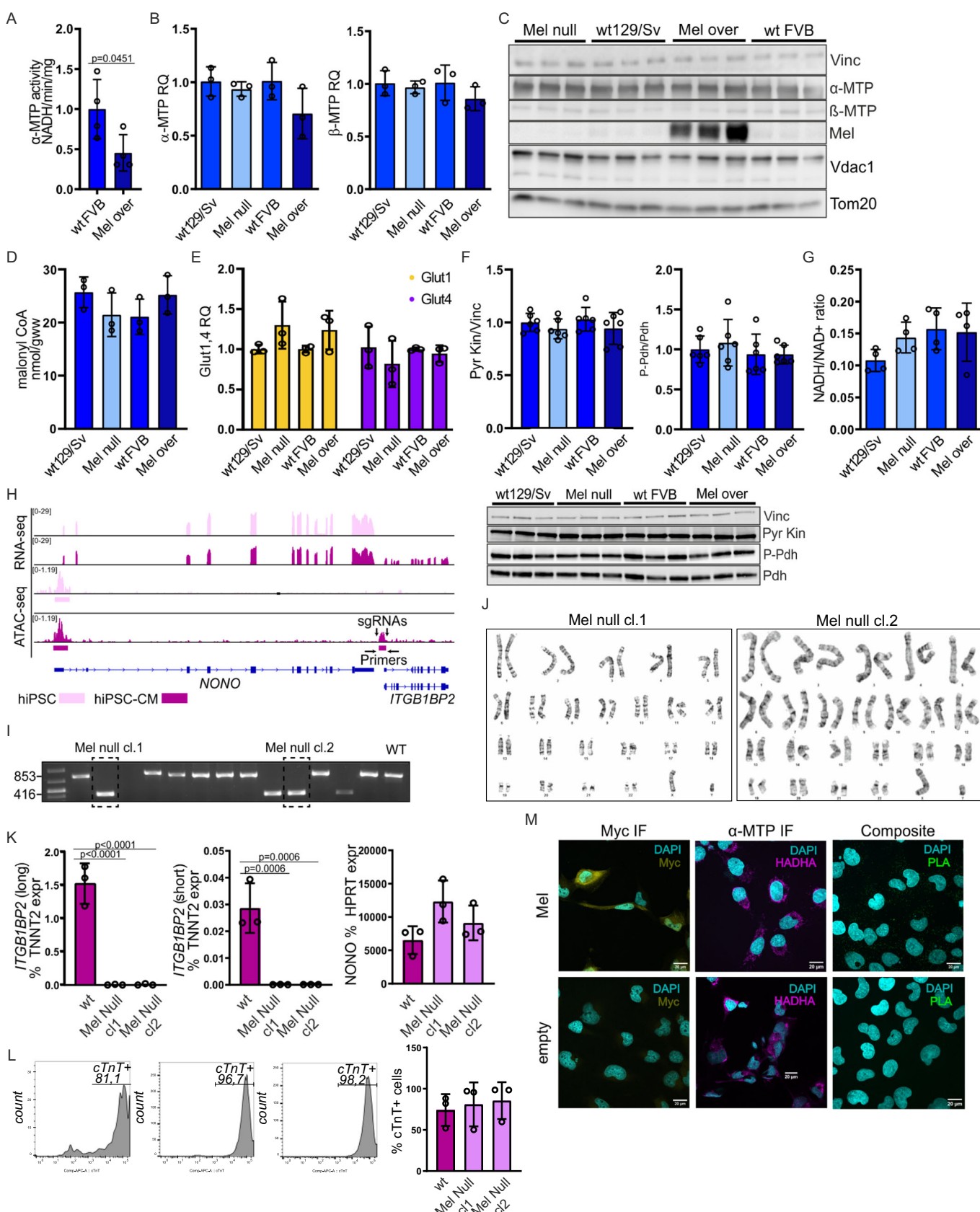

**Figure EV2. Characterization of MTP in mice with different expressions of Melusin and of Melusin-null hiPSCs.**

(A) Modulation of α-MTP activity in wild-type and Mel over mice detected as NADH reduction in time in cardiac homogenates supplemented with acetoacetyl-CoA. ($n = 3$ per group; mean ± SD; $p$ value by unpaired $t$-test; Data normalized on wt129/Sv media = 1). (B) Expression of α-MTP and β-MTP mRNA evaluated by real-time qPCR in hearts from Mel null, Mel over, and wild-type mice. ($n = 3$ per group; mean ± SD; $p$ value by unpaired $t$-test). (C) Protein level of α-MTP, β-MTP, Vdac1, and Tom20 in cardiac extracts from Mel null, Mel over, and wild-type mice. Melusin was stained as a marker of the different genotypes. Vinculin was stained as a loading control. ($n = 3$ per group). (D) Malonyl-CoA level detected by ELISA assay in Mel null, Mel over, and wild-type hearts. ($n = 3$ per group; mean ± SD; $p$ value by unpaired $t$-test). (E) Expression of Glut1 and Glut4 mRNA evaluated by real-time qPCR in hearts Mel null, Mel over, and wild-type mice. ($n = 3$ per group; mean ± SD; $p$ value by unpaired $t$-test). (F) Protein level of pyruvate kinase and phosphorylation ratio of Pdh in cardiac extracts from Mel null, Mel over, and wild-type mice. ($n = 6$ per group; mean ± SD; $p$ value by unpaired $t$-test). (G) NADH/NAD+ ratio estimated by HPLC analysis in cardiac extracts from Mel null, Mel over, and wild-type mice. ($n = 4$ per group; mean ± SD; $p$ value by unpaired $t$-test). (H) RNA-seq and ATAC-seq data of WT hESCs and hESC-CMs. *NONO* expression is constant (RNA-seq data) and the promoter region is accessible in both cell types (ATAC-seq data). *ITGB1BP2* promoter is accessible for transcription only in hiPSC-CMs. The sgRNA guides for Melusin knocking out were designed to bind the promoter region up- and down-stream the ATAC-seq peak. Primers for genotyping were designed against outside of the predicted cut sites. (I) Genotyping of Mel null hiPSC clones. The 853 bp band is the product of WT locus amplification, and the lower band (416 bp) is the result of promoter deletion. WT hiPSCs were used as control. All genome-edited clones show either the WT or promoter deletion band since Melusin is X-linked and the experiment involved male hiPSCs. (J) Karyotypes of Mel null cl.1 and cl.2 hiPSCs. Metaphase chromosomes were stained by G-banding. The karyotypes of Mel null hiPSCs are normal without clonal abnormalities and belong to a male individual. (K) RT-qPCR of WT, Mel null cl.1 and cl.2 hiPSC-CMs. *ITGB1BP2* mRNA was normalized on the muscle-specific marker *TNNT2*, so as to better account for variations in hiPSC-CM purity in samples analyzed prior to lactate selection. Two RefSeq isoforms for the *ITGB1BP2* gene were probed: the full-length canonical Melusin isoform (NM_012278.4) and a shorter isoform proved to be expressed at ~2% of Melusin (NM_001303277.3). The short isoform is predicted to share the same promoter as Melusin. Both isoforms were undetectable in both Mel null clones, validating the knockout strategy. To rule out a negative effect of the *ITGB1BP2* promoter deletion on the housekeeping *NONO* gene, located very close upstream of *ITGB1BP2*, *NONO* mRNA was analyzed and normalized on the housekeeping gene *HPRT*. ($n = 3$ total RNA isolation per group from hiPSC-CMs independently differentiated; mean ± SD; $p$ value by one-way ANOVA with Bonferroni correction). (L) cTnT + cells by flow cytometry of WT, Mel null cl.1 and cl.2 hiPSC-CMs post lactate selection (histograms are gated based on isotype control stains), and relative graph. ($n = 3$ per group; each experiment is performed by using hiPSC-CMs independently differentiated; mean ± SD; $p$ value by one-way ANOVA with Bonferroni correction). (M) Representative Immunofluorescence (IF) and PLA signal of Myc-Melusin (Myc-tag antibody) and α-MTP (HADHA antibody) in AC16 cells empty (empty) or expressing Myc-Melusin (Mel). DAPI was used to stain nuclei. Images were captured by confocal microscopy. Scale bare = 20 μm. Representative of $n = 3$ independent experiments. Vinc vinculin, Vdac1 voltage-dependent anion-selective channel 1, Tom20 translocase of outer membrane 20, Pyr Kin pyruvate kinase, P-Pdh phosphorylated pyruvate dehydrogenase, Pdh pyruvate dehydrogenase, sgRNAs single guide RNAs, *ITGB1BP2* Integrin β1 binding protein 2, bp base pair, *TNNT2* troponin type2, cTNT cardiac troponin T, *HPRT* hypoxanthine phosphoribosyltransferase 1, PLA proximity ligation assay.

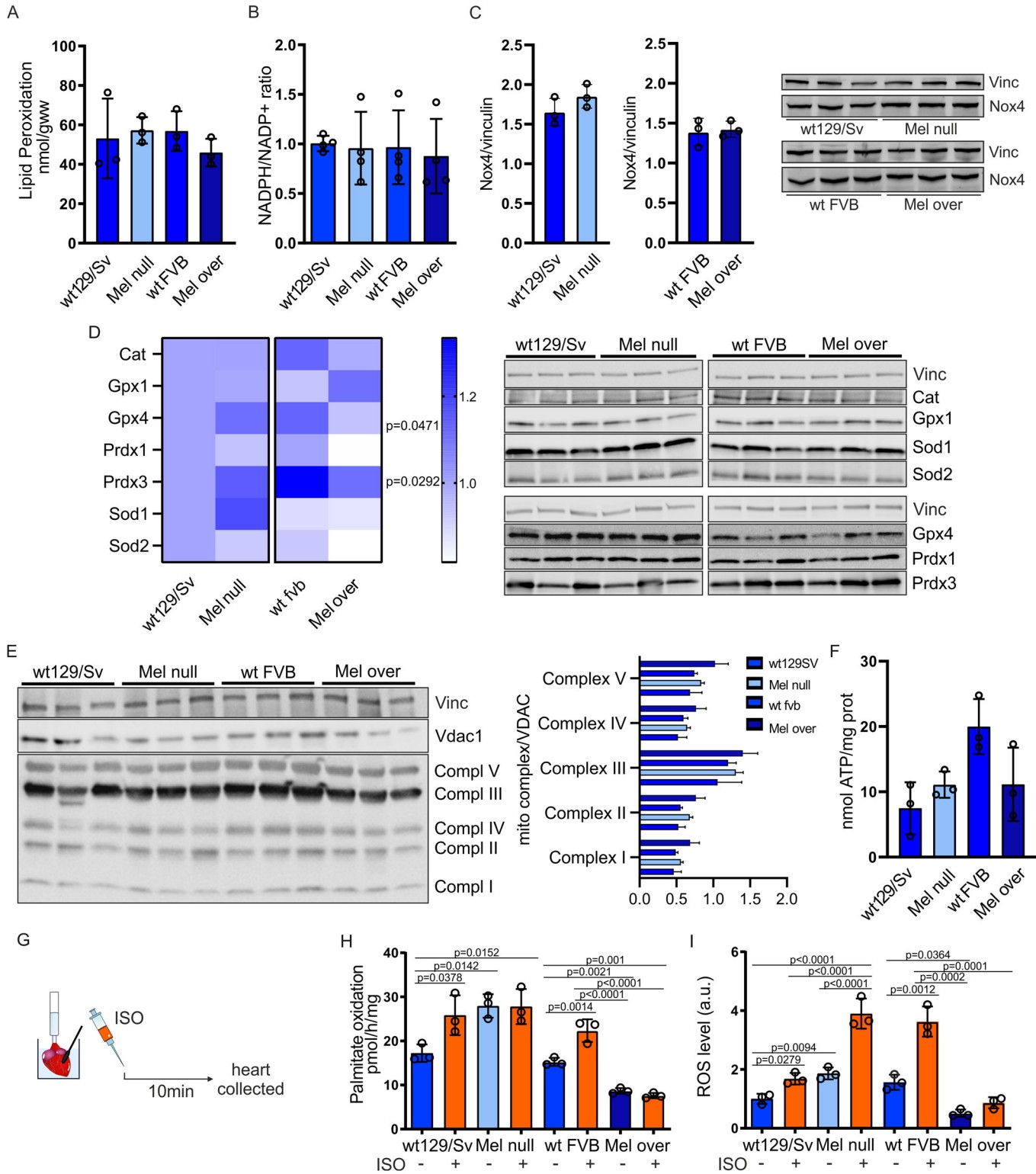

◄ **Figure EV3.** **Determination of parameters impacting mitochondrial function and ROS generation and evaluation of high-fat diet effect.**

(**A**) Modulation of lipid peroxides detected as colorimetric reaction with malonaldehyde and 4-hydroxyalkenal in cardiac extracts of wild-type, Mel null, and Mel over mice. ($n = 3$ per group. mean ± SD; $p$ value by unpaired $t$-test). (**B**) NADPH/NADP+ ratio estimated by HLPC analysis in cardiac extracts from Mel null, Mel over, and wild-type mice. ($n = 4$ per group; mean ± SD; $p$ value by unpaired $t$-test). (**C**) Protein level of Nox4 in cardiac extracts from Mel null, Mel over, and wild-type mice in basal conditions (graphs of relative quantification (left) and immunostaining (right)). Vinculin was stained as a loading control. ($n = 3$ per group; mean ± SD; $p$ value by unpaired $t$-test). (**D**) Heat map (left) and representative immunostaining (right) of the antioxidant enzymes catalase (Cat), glutathione peroxidase 1 (Gpx1), glutathione peroxidase 4 (Gpx4), peroxiredoxin 1 (Prdx 1), peroxiredoxin 3 (Prdx 3), superoxide dismutase 1 (Sod 1), superoxide dismutase 2 (Sod 2) in cardiac extracts from Mel null, Mel over, and wild-type mice. Vinculin was stained as a loading control. ($n = 6$ per group; mean ± SD; $p$ value by unpaired $t$-test). (**E**) Left: Immunostaining for the five mitochondrial complexes in cardiac extracts from Mel null, Mel over, and wild-type mice. Vdac1 was stained as a marker of mitochondrial quantity. Vinculin was stained as a loading control. Right: relative quantifications. ($n = 3$ per group. mean ± SD; $p$ value by unpaired $t$-test). (**F**) Level of ATP in total cardiac extracts of wild-type, Mel null, and Mel over mice, detected by luciferin-luciferase assay. ($n = 3$ per group; mean ± SD; $p$ value by unpaired $t$-test). (**G**) Isoproterenol treatment protocol: hearts from Mel null, Mel over, and wild-type mice were excised and perfused ex-vivo with 1 μM isoproterenol (ISO +) or saline (ISO −), as control. (**H**) Modulation of palmitate oxidation, evaluated as generation of radioactive metabolites of [1−$^{14}$C]-palmitate, in cardiac isolated mitochondria from Mel null, Mel over, and wild-type mice treated as described in (**F**). ($n = 3$ per group; mean ± SD; $p$ value by two-way ANOVA with Tukey correction (performed separately for the two mice strains)). (**I**) ROS content, measured as DCFDA-AM fluorescence, in cardiac isolated mitochondria from Mel null, Mel over, and wild-type mice treated as described in (**A**). Data normalized on wt129/Sv (wt129/Sv media = 1). ($n = 3$ per group; mean ± SD; $p$ value by two-way ANOVA with Tukey correction (performed separately for the two mice strains)). Cat catalase, Gpx1 glutathione peroxidase 1, Gpx4 glutathione peroxidase 4, Prdx 1 peroxiredoxin 1, Prdx 3 peroxiredoxin 3, Sod 1 superoxide dismutase 1, Sod 2 Superoxide dismutase 2, Vinc vinculin, Vdac1 voltage-dependent anion-selective channel 1, ISO isoproterenol.

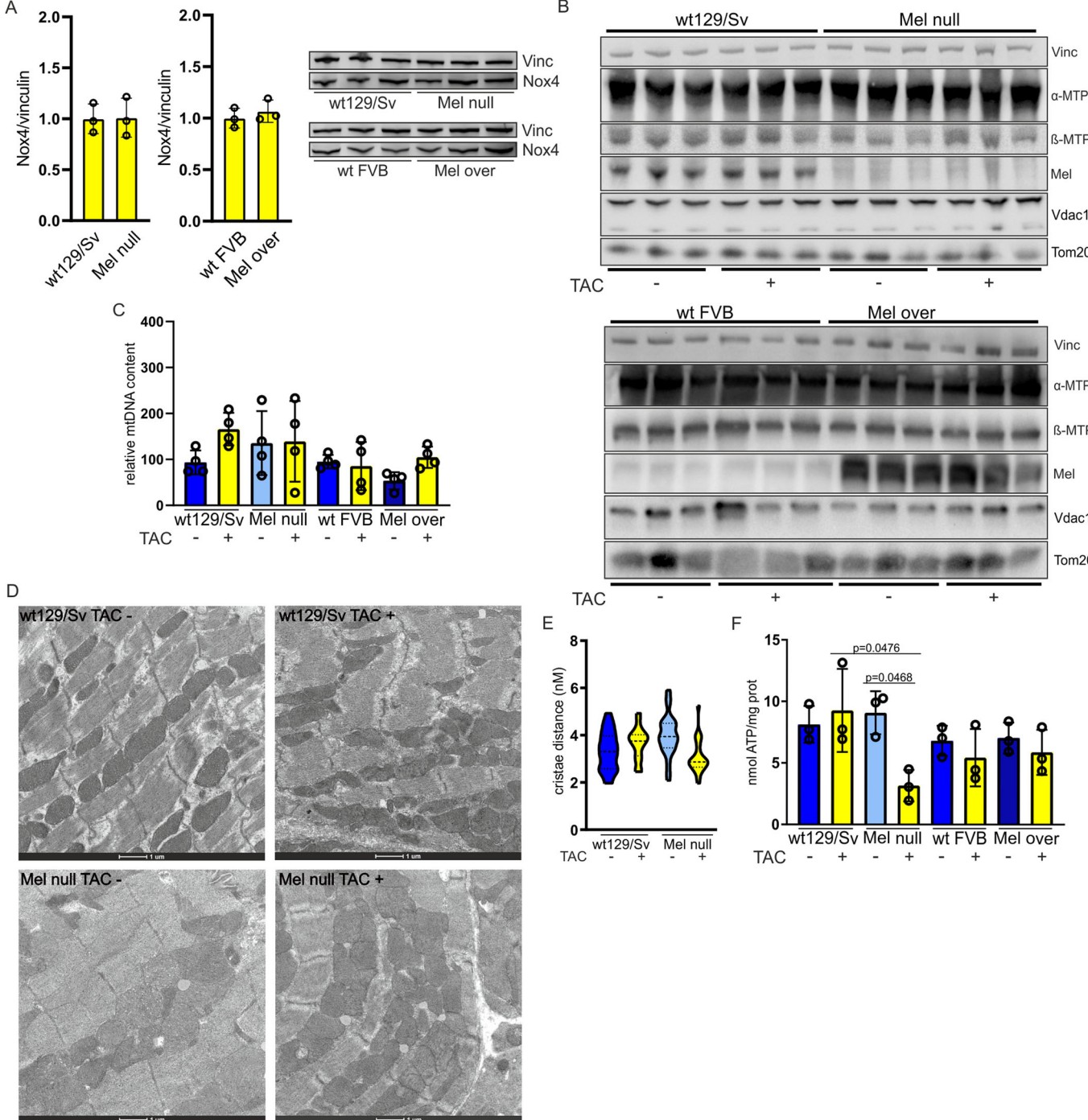

◀

**Figure EV4.  Evaluation of protein expression and structural morphology of mitochondria from mice subjected to pressure overload.**

(A) Protein level of Nox4 in cardiac extracts from Mel null, Mel over, and wild-type mice subjected to TAC surgery for 4 days (graphs of relative quantification (left) and immunostaining (right)). Vinculin was stained as a loading control. ($n = 3$ per group; mean ± SD; $p$ value by unpaired $t$-test). (B) Protein level of α-MTP, β-MTP, Vdac1, and Tom20 in cardiac extracts from Mel null, Mel over, and wild-type mice subjected to TAC (TAC +) or sham (TAC −) surgery for 4 days, as described in 3D. Melusin was stained as a marker of the different genotypes. Vinculin was stained as a loading control. ($n = 3$ per group). (C) Relative mtDNA content compared to genomic DNA evaluated by quantitative real-time PCR analysis in hearts of wild-type, Mel null and Mel over mice subjected to TAC (TAC +) or sham (TAC −) surgery for 4 days, as described in 3D. ($n = 4$ per group; mean ± SD; $p$ value by two-way ANOVA with Tukey correction (performed separately for the two mice strains)). (D) Representative TEM images of cardiac sections of wild-type and Mel null hearts subjected to TAC (TAC +) or sham (TAC −) surgery for 4 days, as described in 3D. Scale bare $= 1$ μm. (E) Relative quantification of cristae distance in nm. Data were means ± SD. ($n = 25$ cristae from mitochondria from three different animals per group. mean ± SD; $p$ value by two-way ANOVA with Bonferroni correction). (F) Level of ATP, detected by luciferin-luciferase assay, in total cardiac extracts of wild-type, Mel null and Mel over mice subjected to TAC (TAC +) or sham (TAC −) surgery for 4 days, as described in 3D. ($n = 3$ per group; mean ± SD; $p$ value by two-way ANOVA with Tukey correction (performed separately for the two mice strains)). Vinc vinculin; Vdac1 voltage-dependent anion-selective channel 1, Tom20 translocase of outer membrane 20, TAC transverse aortic constriction, TEM transmission electron microscopy.

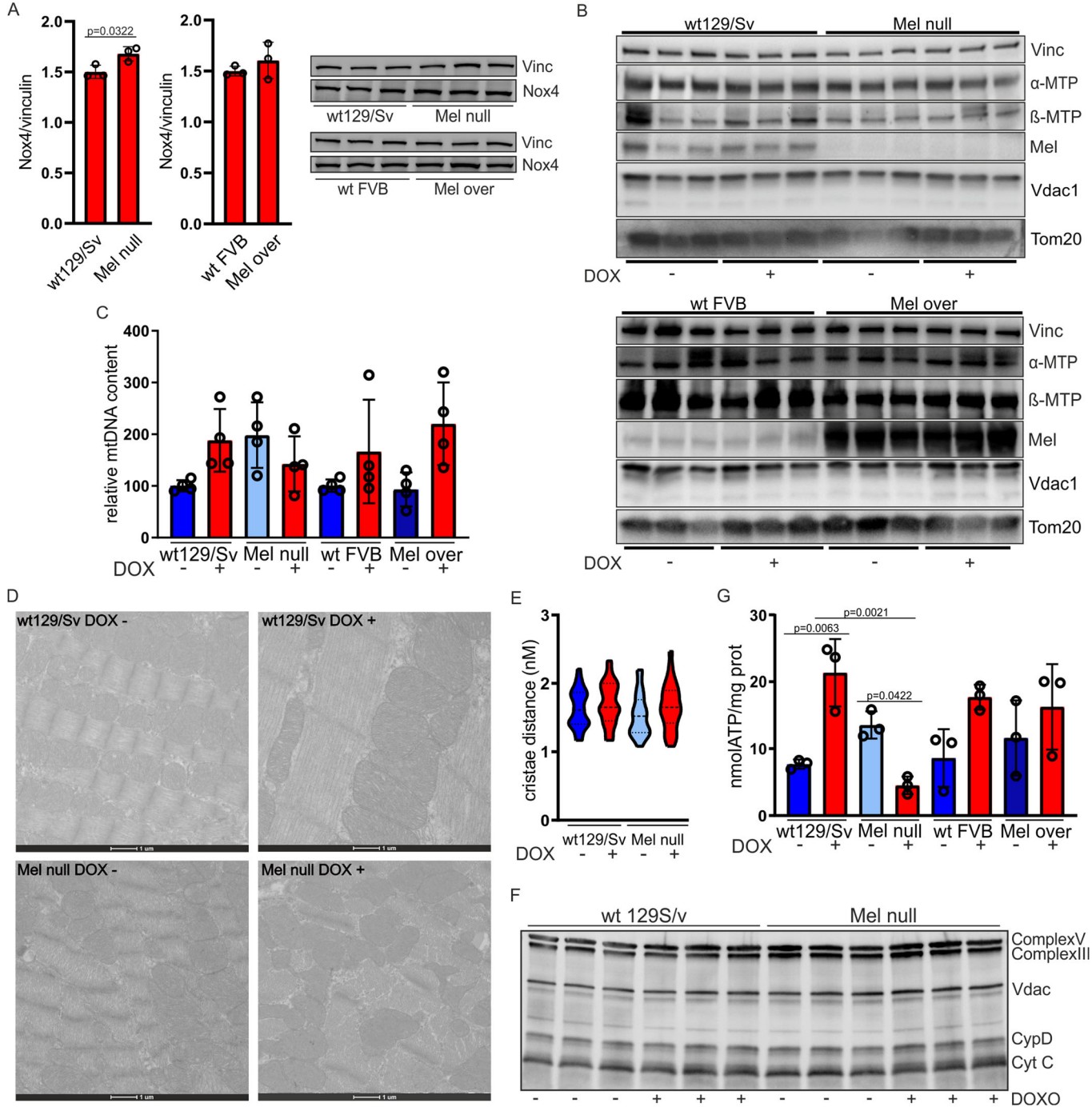

**Figure EV5. Evaluation of protein expression and structural morphology of mitochondria from mice subjected to doxorubicin cardiotoxicity.**

(A) Protein level of Nox4 in cardiac extracts from Mel null, Mel over, and wild-type mice treated with 4 mg/kg doxorubicin for 6 h (graphs of relative quantification (left) and immunostaining (right)). Vinculin was stained as a loading control. ($n = 3$ per group; mean ± SD; $p$ value by unpaired $t$-test). (B) Protein level of α-MTP, β-MTP, Vdac1, and Tom20 in cardiac extracts from Mel null, Mel over, and wild-type mice treated with 4 mg/kg doxorubicin (DOX +) or saline (DOX −) for 6 h, as described in 3F. Melusin was stained as a marker of the different genotypes. Vinculin was stained as a loading control. ($n = 3$ per group). (C) Relative mtDNA content compared to genomic DNA evaluated by quantitative real-time PCR analysis in hearts of wild-type, Mel null, and Mel over mice treated with 4 mg/kg doxorubicin (DOX +) or saline (DOX −) for 6 h, as described in 3F. ($n = 4$ per group; mean ± SD; $p$ value by two-way ANOVA with Tukey correction (performed separately for the two mice strains)). (D) Representative TEM images of cardiac sections of wild-type and Mel null mice treated with 4 mg/kg doxorubicin (DOX +) or saline (DOX −) for 6 h, as described in 3F. Scale bar = 1 μm. (E) Relative quantification of cristae distance in nm. Data were means ± SD. ($n = 25$ cristae from mitochondria from three different animals per group. mean ± SD; $p$ value by two-way ANOVA with Bonferroni correction). (F) Integrity of isolated mitochondria from wild-type 129S/v and Mel null hearts, untreated or treated with doxorubicin, estimated as immunostaining of specific markers of different mitochondrial compartments (Complex Va, Complex III for the inner membrane, Vdac for the outer membrane, cyclophilin D for the matrix, cytochrome C for the intermembrane space. ($n = 3$ per group). (G) Level of ATP, detected by luciferin-luciferase assay, in total cardiac extracts of wild-type, Mel null and Mel over mice treated with 4 mg/kg doxorubicin (DOX +) or saline (DOX −) for 6 h, as described in 3F. ($n = 3$ per group; mean ± SD; $p$ value by two-way ANOVA with Tukey correction (performed separately for the two mice strains)). Vinc vinculin, Vdac voltage-dependent anion-selective channel, Tom20 translocase of outer membrane 20, DOX doxorubicin, TEM transmission electron microscopy, CypD cyclophilin D, Cyt C cytochrome C.

