## [Peer Review File · EMBO Molecular Medicine]

An intrinsic mechanism of metabolic tuning promotes cardiac resilience to stress

Matteo Sorge, Giulia Savorè, Andrea Gallo, Davide Acquarone, Mauro Sbroggiò, Silvia Velasco, Federica Zamporlini, Saveria Femminò, Enrico Moiso, Giampaolo Morciano, Elisa Balmas, Andrea Raimondi, Gabrielle Nattenberg, Rachele Stefania, Carlo Tacchetti, Angela Rizzo, Paola Corsetto, Alessandra Ghigo, Emilia Turco, Fiorella Altruda, Lorenzo Silengo, Paolo Pinton, Nadia Raffaelli, Nathan Sniadecki, Claudia Penna, Pasquale Pagliaro, Emilio Hirsch, Chiara Riganti, Guido Tarone, Alessandro Bertero, and Mara Brancaccio

Corresponding authors: Matteo Sorge (matteo.sorge@unito.it) , Mara Brancaccio (mara.brancaccio@unito.it)

Review Timeline:

Submission Date:	22nd May 24
Editorial Decision:	10th Jun 24
Revision Received:	19th Jul 24
Editorial Decision:	31st Jul 24
Revision Received:	8th Aug 24
Accepted:	9th Aug 24

Editor: Lise Roth

Transaction Report:

(Note: Please note that the manuscript was previously reviewed at another journal and the reports were taken into account in the decision making process at EMBO Molecular Medicine. Since the original reviews are not subject to EMBO Press' transparent review process policy, the reports and author response cannot be published. With the exception of the correction of typographical or spelling errors that could be a source of ambiguity, letters and reports are not edited. Depending on transfer agreements, referee reports obtained elsewhere may or may not be included in this compilation. Referee reports are anonymous unless the Referee chooses to sign their reports.)

10th Jun 2024

Dear Dr. Sorge,

Thank you for the submission of your manuscript to EMBO Molecular Medicine, following rejection post-revision at a different journal.

Briefly, the initial referees acknowledged the work that had been done during the revisions, however they also mentioned remaining concerns regarding the assessment of fatty acid oxidation, ROS measurements and glucose uptake.

We have sent your newly revised manuscript to a single reviewer for evaluation, together with the existing referees' reports, and your point-by-point rebuttal letters. We have now received the enclosed report from this reviewer, who is positive and supportive of publication pending minor revisions, that mostly entail discussing/toning down some of the claims.

We will therefore welcome the submission of your revised manuscript according to this referee's recommendations.

EMBO Molecular Medicine encourages a single round of revision only and therefore, acceptance or rejection of the manuscript will depend on the completeness of your responses included in the next, final version of the manuscript. For this reason, and to save you from any frustrations in the end, I would strongly advise against returning an incomplete revision.

We require:

4) A .docx formatted letter INCLUDING the reviewers' reports and your detailed point-by-point responses to their comments. As part of the EMBO Press transparent editorial process, the point-by-point response is part of the Review Process File (RPF), which will be published alongside your paper.

5) A complete author checklist, which you can download from our author guidelines (<https://www.embopress.org/page/journal/17574684/authorguide#submissionofrevisions>). Please insert information in the checklist that is also reflected in the manuscript. The completed author checklist will also be part of the RPF.

6) Please note that all corresponding authors are required to supply an ORCID ID for their name upon submission of a revised manuscript.

7) It is mandatory to include a 'Data Availability' section after the Materials and Methods. Before submitting your revision, primary datasets produced in this study need to be deposited in an appropriate public database, and the accession numbers and database listed under 'Data Availability'. Please remember to provide a reviewer password if the datasets are not yet public (see <https://www.embopress.org/page/journal/17574684/authorguide#dataavailability>).

In case you have no data that requires deposition in a public database, please state so in this section ("This study includes no data deposited in external repositories.").

Note that the Data Availability Section is restricted to new primary data that are part of this study.

8) For data quantification: please specify the name of the statistical test used to generate error bars and P values, the number (n) of independent experiments (specify technical or biological replicates) underlying each data point and the test used to calculate p-values in each figure legend. The figure legends should contain a basic description of n, P and the test applied. Graphs must include a description of the bars and the error bars (s.d., s.e.m.). Please provide exact p values.

9) Our journal encourages inclusion of *data citations in the reference list* to directly cite datasets that were re-used and obtained from public databases. Data citations in the article text are distinct from normal bibliographical citations and should directly link to the database records from which the data can be accessed. In the main text, data citations are formatted as

follows: "Data ref: Smith et al, 2001" or "Data ref: NCBI Sequence Read Archive PRJNA342805, 2017". In the Reference list, data citations must be labeled with "[DATASET]". A data reference must provide the database name, accession number/identifiers and a resolvable link to the landing page from which the data can be accessed at the end of the reference. Further instructions are available at .

13) Author contributions: CRediT has replaced the traditional author contributions section because it offers a systematic machine readable author contributions format that allows for more effective research assessment. Please remove the Authors Contributions from the manuscript and use the free text boxes beneath each contributing author's name in our system to add specific details on the author's contribution. More information is available in our guide to authors.

16) As part of the EMBO Publications transparent editorial process initiative (see our Editorial at <http://embomolmed.embopress.org/content/2/9/329>), EMBO Molecular Medicine will publish online a Review Process File (RPF) to accompany accepted manuscripts.

In the event of acceptance, this file will be published in conjunction with your paper and will include the anonymous referee reports, your point-by-point response and all pertinent correspondence relating to the manuscript. Let us know whether you agree with the publication of the RPF and as here, if you want to remove or not any figures from it prior to publication. Please note that the Authors checklist will be published at the end of the RPF.

I look forward to receiving your revised manuscript.

Yours sincerely,

Lise Roth

***** Reviewer's comments *****

Referee #1 (Comments on Novelty/Model System for Author):

The method they used to assess FA oxidation capacity is not optimal as it actually measures FA oxidation in mitochondria isolated from hearts thus lacking the interplay with either the cellular environment or the adaptation to cardiac contractility changes. Having said that, even this analysis provides some useful information that may not be extrapolated to the real situation. The authors have added a relevant clause in the "Limitations" paragraph of the Discussion section.

Referee #1 (Remarks for Author):

In this manuscript, Matteo Sorge et al. studied the role of Melusin, a muscle-specific chaperone, in controlling cardiac fatty acid oxidation (FAO) and modulating oxidative stress. Their study encompasses various animal models of heart failure and metabolic analyses in mitochondria isolated from hearts or hiPSC cultures. They have produced ample data and propose that Melusin is an inhibitor of FAO and oxidative stress, which is supported by their data. On the other hand, they attribute oxidative stress to increased FAO, which is the result of partial assessment of the metabolic pathways that may be involved. To this end, they propose that activation of Melusin can be used as a beneficial intervention for treating heart failure with increased afterload or doxorubicin-induced cardiomyopathy. Here are my major and minor comments:

1. Attributing oxidative stress in increased FAO is an exaggerating interpretation of their findings. They seem to either ignore or underestimate the role of other important factors that contribute to the cellular redox balance. With fatty acids being the primary cardiac fuel in healthy hearts, any shift towards lower utilization of fatty acids in favor of glucose use for oxidative metabolism and ATP synthesis cannot be seen unilaterally. Unless they study glucose fate when mitochondrial FAO is induced upon melusin ablation, the statement that FAO stimulates ROS formation and oxidative stress is not a conclusion that is safely deprived from possible misinterpretation. The authors need to either study or discuss thoroughly what may happen to glucose when FAO is blocked and how deviation of glucose from pathways that it ends up to when FAO is the primary source of ATP may affect ROS formation. Several glucose catabolism pathways, such as the polyol pathway and the hexosamine biosynthesis pathway have been associated with oxidative stress. A glycolysis derivative, Serine, has been also associated with anti-oxidant effects. It is highly likely that inhibition of FAO may channel glucose almost exclusively towards mitochondrial oxidation, which will eventually suppress pathways that are still Glucose-dependent and contribute to ROS formation. Especially in conditions like pressure overload that are accompanied by stimulation of Akt and increased glucose uptake, increased availability of glucose for pro-oxidant pathways is highly probable to be the case.
2. Lower number of mitochondria that they report in the present study may be the reason for oxidative stress, as this not an unusual outcome of "overwhelmed" mitochondria that contribute to covering the vast ATP needs of the cardiomyocytes. Treatment of cells with resveratrol should increase mitochondrial abundance and would help to test if this would alleviate the detrimental effects of melusin ablation.
3. A cardiac oxidative stress study would be more complete with assessment of the expression of pro-oxidant proteins, such as NADPH oxidases.
4. How does doxorubicin treatment affect melusin expression?
5. In the Introduction and the Discussion sections, they refer to some studies that presumably support their statement about fatty acid oxidation being a more potent oxidative stress inducer than glucose. The authors need to read more carefully those studies. To the best of my understanding, the Chen et al study (JBC, 2003) does not contain the term glucose even once. It is not clear where the Anderson et al study (JCI, 2009) compares fatty acids and glucose as potential inducers of oxidative stress. The data of the Seifert et al study (JBC, 2010) are not relevant at all to the authors' statement. Even the studies that have compared the capacity of pyruvate and fatty acids in promoting H₂O₂ production need to be examined very carefully for stoichiometry of the used substrates in relation to provided oxygen before a safe conclusion about fatty acids being a more potent inducer of oxidative stress.
6. Figure EV2F: It is unclear why the Western blotting images are not included.
7. The text includes some typos and grammatical errors.

Point-by-point discussion to Reviewer's requests.

Reviewers' comments are in black, while our responses are in blue. The main modifications made to the manuscript are in magenta. References to pages, lines and images are referred to the new version of the manuscript.

Referee #1

1. Attributing oxidative stress in increased FAO is an exaggerating interpretation of their findings. They seem to either ignore or underestimate the role of other important factors that contribute to the cellular redox balance. With fatty acids being the primary cardiac fuel in healthy hearts, any shift towards lower utilization of fatty acids in favor of glucose use for oxidative metabolism and ATP synthesis cannot be seen unilaterally. Unless they study glucose fate when mitochondrial FAO is induced upon Melusin ablation, the statement that FAO stimulates ROS formation and oxidative stress is not a conclusion that is safely deprived from possible misinterpretation. The authors need to either study or discuss thoroughly what may happen to glucose when FAO is blocked and how deviation of glucose from pathways that it ends up to when FAO is the primary source of ATP may affect ROS formation. Several glucose catabolism pathways, such as the polyol pathway and the hexosamine biosynthesis pathway have been associated with oxidative stress. A glycolysis derivative, Serine, has been also associated with anti-oxidant effects. It is highly likely that inhibition of FAO may channel glucose almost exclusively towards mitochondrial oxidation, which will eventually suppress pathways that are still Glucose-dependent and contribute to ROS formation. Especially in conditions like pressure overload that are accompanied by stimulation of Akt and increased glucose uptake, increased availability of glucose for pro-oxidant pathways is highly probable to be the case.

We thank the Reviewer for this important suggestion. In the revised version of the manuscript, we discussed the role of glucose in the generation of ROS in the myocardium, in particular when lipids are the main substrates for mitochondrial oxidation (Introduction Page 2 lines 70-76; Discussion Page 13 lines 355-362; reported below).

Introduction:

“The cardiac metabolic preference for FAO, as defined by the Randle cycle (Hue & Taegtmeyer, 2009), assures that, when available, the myocardium preferentially oxidizes lipids instead of glucose to produce more ATP. However, this comes at the cost of increased ROS production. FAO generates ROS (Isei et al, 2022; Speijer et al, 2014; St-Pierre et al, 2002) and additional oxidative stress arises from glucose that, not being oxidized in the mitochondria, is directed toward other catabolic pathways, such as the polyol pathway and the hexosamine biosynthesis pathway (Battault et al, 2020; Mylonas et al, 2023). In addition, stressful situations, such as pressure overload (Nickel et al, 2015; Suetomi et al, 2018; Wang et al, 2018), ischemia/reperfusion (Bugger & Pfeil, 2020) or doxorubicin-induced cardiotoxicity (Berthiaume & Wallace, 2007), generate oxidative stress that further raise

myocardial ROS levels above a certain threshold, triggering a self-amplification mechanism known as ROS-induced ROS release (RIRR) (Dey et al, 2018; Leach et al, 2001; Tse et al, 2016; Tullio et al, 2013; Zorov et al, 2000)."

Discussion:

"Moreover, intermediates of fatty acid catabolism may inhibit the electron transport chain, favoring electron escape, and interfere with ROS scavenging (Seifert et al, 2010; St-Pierre et al., 2002). Furthermore, the preference for FAO may redirect glucose toward catabolic pathways different from mitochondrial oxidation, such as the polyol pathway and the hexosamine biosynthesis pathway, known to generate oxidative stress (Battault et al., 2020; Mylonas et al., 2023). In addition, the reduction of glycolysis decreases the production of serine, which has anti-oxidant effects (He et al, 2023; Mylonas et al, 2023)."

2. Lower number of mitochondria that they report in the present study may be the reason for oxidative stress, as this not an unusual outcome of "overwhelmed" mitochondria that contribute to covering the vast ATP needs of the cardiomyocytes. Treatment of cells with resveratrol should increase mitochondrial abundance and would help to test if this would alleviate the detrimental effects of Melusin ablation.

We previously deduced that the number of mitochondria in the hearts of mice of different genotypes, in basal conditions and after challenge, is not significantly different by analyzing the mitochondrial proteins VDAC1 and TOM20 by Western blot (see Figure EV2, EV4 and EV5). To strengthen this result, we now evaluated the mitochondrial DNA content by qPCR on mouse hearts and demonstrated that no significant differences are present among the various mouse genotypes and conditions (see Figure EV4 and EV5). The results are shown below for the benefit of the Reviewer.

Relative mtDNA content compared to genomic DNA evaluated by quantitative real-time PCR analysis in hearts of wild-type, Mel null and Mel over mice subjected to TAC (TAC +) or sham (TAC -) surgery for 4 days (A) or treated with 4 mg/kg doxorubicin (DOX +) or saline (DOX -) for 6 hours (B).

3. A cardiac oxidative stress study would be more complete with assessment of the expression of pro-oxidant proteins, such as NADPH oxidases.

In our study we described a specific regulation of mitochondrial lipid metabolism and ROS generation. The constitutively active NADPH oxidase 4 is one of the main isoforms expressed in the heart and has been described to localize in the mitochondria (Kuroda *et al*, 2010; Shanmugasundaram *et al*, 2017) and regulate lipid and glucose metabolism (Nabeebaccus *et al*, 2023). As suggested by the Reviewer, we analyzed the Nox4 protein expression level in the heart of wild type, Mel null and Mel over-expressing mice in basal conditions (Figure EV3) and after challenge (Figure EV4 and EV5). No differences were detected in basal conditions and 4-days after TAC among genotypes. We only found a mild but significant increase in Nox4 level in Mel null mice 6 hours after doxorubicin treatment. Given that Nox4 expression is known to be induced by ROS (Li *et al*, 2023; Yiu *et al*, 2016), this is likely due to higher ROS content in Mel null mice compared to wild types. These results are shown below for the benefit of the Reviewer.

Protein level of Nox4 in cardiac extracts from Mel null, Mel over and wild-type mice in basal conditions (A), subjected to TAC surgery for 4 days (B) and treated with 4 mg/kg doxorubicin for 6 hours (C). Vinculin was stained as loading control.

4. How does doxorubicin treatment affect melusin expression?

We analyzed Melusin expression levels in the heart of mice 6 hours after doxorubicin injection and we did not find significant differences in respect to untreated mice, as shown in Figure EV5. Similar results have been obtained in mice 5 weeks after a multiple dose treatment with doxorubicin (see figure below).

Protein level of Melusin in cardiac extracts from mice 5 weeks after treatment with 3 doses of doxorubicin (4 mg/kg at days 0, 7, and 14) or saline as control. Vinculin was stained as loading control. Desmin and Troponin were stained as additional markers of cardiomyocytes (A). Graphs of relative quantifications (B).

5. In the Introduction and the Discussion sections, they refer to some studies that presumably support their statement about fatty acid oxidation being a more potent oxidative stress inducer than glucose. The authors need to read more carefully those studies. To the best of my understanding, the Chen et al study (JBC, 2003) does not contain the term glucose even once. It is not clear where the Anderson et al study (JCI, 2009) compares fatty acids and glucose as potential inducers of oxidative stress. The data of the Seifert et al study (JBC, 2010) are not relevant at all to the authors' statement. Even the studies that have compared the capacity of pyruvate and fatty acids in promoting H₂O₂ production need to be examined very carefully for stoichiometry of the used substrates in relation to provided oxygen before a safe conclusion about fatty acids being a more potent inducer of oxidative stress.

We apologize for the inaccuracy. We re-analyzed the references as suggested by the Reviewer and we reformulated our statements and partially revised references (Results page 6 lines 186-188; Discussion page 13 lines 347-348).

6. Figure EV2F: It is unclear why the Western blotting images are not included.

As suggested, we added the Western blotting images in Figure EV2F.

7. The text includes some typos and grammatical errors.

We revised the manuscript and corrected the mistakes.

He L, Ding Y, Zhou X, Li T, Yin Y (2023) Serine signaling governs metabolic homeostasis and health. *Trends Endocrinol Metab* 34: 361-372

Kuroda J, Ago T, Matsushima S, Zhai P, Schneider MD, Sadoshima J (2010) NADPH oxidase 4 (Nox4) is a major source of oxidative stress in the failing heart. *Proc Natl Acad Sci U S A* 107: 15565-15570

Li L, Lu M, Peng Y, Huang J, Tang X, Chen J, Li J, Hong X, He M, Fu H *et al* (2023) Oxidatively stressed extracellular microenvironment drives fibroblast activation and kidney fibrosis. *Redox Biol* 67: 102868

Mylonas N, Drosatos K, Mia S (2023) The role of glucose in cardiac physiology and pathophysiology. *Curr Opin Clin Nutr Metab Care* 26: 323-329

Nabeebaccus AA, Reumiller CM, Shen J, Zoccarato A, Santos CXC, Shah AM (2023) The regulation of cardiac intermediary metabolism by NADPH oxidases. *Cardiovasc Res* 118: 3305-3319

Shanmugasundaram K, Nayak BK, Friedrichs WE, Kaushik D, Rodriguez R, Block K (2017) NOX4 functions as a mitochondrial energetic sensor coupling cancer metabolic reprogramming to drug resistance. *Nat Commun* 8: 997

Yiu WH, Wong DW, Wu HJ, Li RX, Yam I, Chan LY, Leung JC, Lan HY, Lai KN, Tang SC (2016) Kallistatin protects against diabetic nephropathy in db/db mice by suppressing AGE-RAGE-induced oxidative stress. *Kidney Int* 89: 386-398

31st Jul 2024

Dear Dr. Sorge,

Thank you for submitting your revised study. We have now received the feedback from referee #1, who is satisfied with the revisions. I will therefore be able to accept your manuscript once the following editorial points will be addressed:

1/ Manuscript text:

- Please note that emails bounced for Guido Tarone (guido.tarone@unito.it) and moiso3@mskcc.org.
- We note that you currently have together with you a total of 3 co-corresponding authors. Is that correct? Do you confirm equal contribution of these 3 people, able to take full responsibility for the paper and its content?
- Please remove the colored font and only keep in track changes mode any new modification.
- Methods:
 - o All Materials and Methods need to be described in the main text using our 'Structured Methods' format, which is now required for all research articles. According to this format, the Methods section includes a Reagents and Tools Table (listing key reagents, experimental models, software and relevant equipment and including their sources and relevant identifiers) followed by a Methods and Protocols section describing the methods using a step-by-step protocol format. A downloadable template (.docx) for the Reagents and Tools Table can be found in our author guidelines: <https://www.embopress.org/page/journal/17574684/authorguide#structuredmethods>
 - o Antibodies: please make sure that dilutions/concentrations used are provided.
- Data Availability should be placed at the end of Methods.
- Funding should be merged with Acknowledgements.
- "Competing interests" should be renamed "Disclosure statement and competing interests".
- Figure legends should be moved after the References; expanded view figures should be renamed "Figure EV1" etc. and grouped in a section after the main figure legends, under the heading "Expanded View Figure Legends"

2/ Figures:

- Please address the queries from our data editors in the figure legends (if not yet addressed):
 1. Please note that the exact p values are not provided in the legends of figures 1a-b; 2d, f, l; 3b-e, g-j; 4b-c; 5b, e-f, l; EV 2k; EV 3h-i.
 2. Please indicate the statistical test used for data analysis in the legend of figure EV 5a.
 3. Please note that information related to n is missing in the legends of figures EV 3c; EV 4a; EV 5a.
 4. Although 'n' is provided, please describe the nature of entity for 'n' in the legends of figures 1l; 2g-i; 4b-c; EV 2k-l; EV 5e.
 5. Please note that the error bars are not defined in the legends of figures EV 3c; EV 4a; EV 5a.
 6. Please note that in figures 1g; EV 2m; EV 4d; EV 5d; the scale bar unit should be corrected from μM to μm (in the figure legend).
- Please mention in the figure legends any re-use of blots (i.e. Figure EV3C and Figure EV4A, Figure 1L)

3/ Synopsis:

Thank you for providing a nice synopsis figure. Please resize it to 550 px wide x 300-600 px high and make sure the text remains legible, ideally with a white background.

4/ As part of the EMBO Publications transparent editorial process initiative (see our Editorial at <http://embomolmed.embopress.org/content/2/9/329>), EMBO Molecular Medicine will publish online a Review Process File (RPF) to accompany accepted manuscripts.

This file will be published in conjunction with your paper and will include the anonymous referee reports, your point-by-point response and all pertinent correspondence relating to the manuscript. Let us know whether you agree with the publication of the RPF and as here, if you want to remove or not any figures from it prior to publication. Please note that the Authors checklist will be published at the end of the RPF.

I look forward to receiving your revised manuscript.

With kind regards,

Lise

***** Reviewer's comments *****

Referee #1 (Remarks for Author):

The authors have addressed my concerns satisfactorily.

The authors addressed the minor editorial issues.

9th Aug 2024

Dear Dr. Sorge,

Thank you for submitting your revised files. I am pleased to inform you that your manuscript is accepted for publication and is now being sent to our publisher to be included in the next available issue of EMBO Molecular Medicine!

The Review Process File will be sent to you for approval.

With kind regards,

Lise Roth
